# Temporal Variational Implicit Neural Representations

## Abstract

We introduce Temporal Variational Implicit Neural Representations (TV-INRs), a probabilistic framework for modeling irregular multivariate time series that enables efficient and accurate individualized imputation and forecasting. By integrating implicit neural representations with latent variable models, TV-INRs learn distributions over time-continuous generator functions conditioned on signal-specific covariates. Unlike existing approaches that require extensive training, fine-tuning or meta-learning, our method achieves accurate individualized predictions through a single forward pass. Our experiments demonstrate that with a single TV-INRs instance, we can accurately solve diverse imputation and forecasting tasks, offering a computationally efficient and scalable solution for real-world applications. TV-INRs excel especially in low-data regimes, where it outperforms existing imputation methods by an order of magnitude in mean squared error.

## 1 Introduction

Time series are a key way to represent data in many domains, from energy consumption to finance, and they frequently contain missing values and irregularities due to sensor malfunctions, collection errors, or resource constraints (Che et al., 2018; Du et al., 2023; Proietti & Pedregal, 2023). These challenges are particularly pronounced in clinical datasets, which often exhibit extreme sparsity (80-90% missingness) and noisy, irregular sampling due to human involvement in non-automated measurements (Silva et al., 2012). In order to impute missing values and forecast future time points, effective solutions must handle these challenges while utilizing available covariates to capture unique temporal dynamics.

Current methods relying on Recurrent Neural Networks (RNNs) (Chung et al., 2015; Che et al., 2018) and Transformers (Bansal et al., 2023; Liu et al., 2023) are generally tailored for regular, dense time series data and require placeholders for missing observations. They also operate in discrete time, and careful design is necessary for continuous time settings (Chen et al., 2024). Alternatively, there exist continuous time series models which use Implicit Neural Representations (INRs) (Sitzmann et al., 2020) to handle irregular time series data (Naour et al., 2024; Cho et al., 2024). By learning a unique continuous function to represent each time series, INRs have great potential for individualization by capturing the unique activity patterns of each subject. However, existing approaches are inflexible, and often require training multiple models, fine-tuning, or meta-learning to handle variations in data availability, prediction length, and individualization. For example, the method presented in Naour et al. (2024) requires the training of separate models for different missingness ratios or horizon lengths, and performs gradient-based meta-learning during inference, resulting in a data-hungry model. Such approaches are impractical in real-world applications where scalability and generalization are crucial, as computational resources may be limited during deployment.

To address these shortcomings, we introduce Temporal Variational Implicit Neural Representations (TV-INRs), a novel probabilistic model for multivariate time series with INRs. We use INRs as generator functions for continuous time series modeling, effectively handling the challenge of irregular sampling. By also integrating latent variable models and amortized variational inference, TV-INRs learns distributions over INRs conditioned on individual signals and their covariates through a learned latent space. This approach is therefore scenario and sample agnostic, accommodating varying levels of missingness or time series length and eliminating the need for task-specific retraining or per-sample optimization. In short, we preserve the benefits of INRs for time series while making them scalable and efficient.

Our model pushes forward multivariate time series analysis with several key contributions:

- We introduce the first fully probabilistic model for multivariate time series using INRs.
- TV-INRs achieves competitive accuracy to gradient-based meta-learning approaches and improves imputation performance in low-data scenarios, without requiring per-sample optimization during inference.
- We demonstrate successful generalization across multiple data settings, including missingness and forecasting horizon length, with a single training. This significantly reduces training requirements relative to comparable models.
- Our results show that the inclusion of covariates enables effective individualization and further increases our model's accuracy with sparse data, demonstrating suitability for real-world applications with extreme missingness, such as healthcare.

## 2 BACKGROUND & RELATED WORK

### 2.1 LEARNING IMPLICIT NEURAL REPRESENTATIONS

**Hypernetworks** denoted as $g_\phi$, are neural networks that generate parameters $\theta = g_\phi(\cdot)$ for another neural network $f_\theta(\cdot)$ (Ha et al., 2016). Hypernetworks can generate task-specific model parameters, making them suitable for meta-learning scenarios that require quick adaptation to new tasks. Zhao et al. (2020) showed that meta-learning a hypernetwork effectively modulates inner-loop optimization and adapts features task-dependently using model-agnostic meta-learning. Nguyen et al. (2022) proposed to generate parameters of the approximate posterior and likelihood of a Variational Autoencoder (VAE) model to perform multiple tasks. Recent works have shown hypernetworks to be useful for generating parameters for implicit neural representations (Dupont et al., 2021; Koyuncu et al., 2023).

**Implicit neural representations (INRs)** offer a novel approach to data representation and modeling complex continuous signals using the weight space (Sitzmann et al., 2020). This formulation is supported by strong theoretical guarantees and makes the model inherently resolution-agnostic and robust to irregular sampling (Sitzmann et al., 2020). By leveraging neural networks, particularly multi-layer perceptrons (MLPs), represented as $f_\theta(\cdot)$, INRs effectively map coordinates to features like color, occupancy, or amplitude. Therefore INRs enable continuous representation of high-dimensional data, offering significant advantages in various domains, including images, 3D shape modeling, spatio-temporal data (Dupont et al., 2021; 2022a; Koyuncu et al., 2023; Park et al., 2024) and geometric structures (Vetsch et al., 2022; Niemeyer et al., 2022), because predictions are not constrained by input range or resolution. Recent works are actively exploring parameterization strategies for INRs. For example, approaches by Dupont et al. (2022b); Strümpler et al. (2022) have used compressed representations of the data as inputs to hypernetworks $g_\phi$, which then generate weights $\theta$ of the INRs $f_\theta(\cdot)$. Peis et al. (2025) uses latent diffusion models to generate a latent variable model to model the weights of INRs via a transformer network. And Park et al. (2024) proposed to learn sample-specific dynamic positional embeddings, rather than modeling INRs weights.

**Meta-learning** is a learning approach where algorithms are designed to improve their learning efficiency and adaptability across different tasks and domain shifts. In model-agnostic meta-learning (MAML), the aim is to fine-tune the trained model using test instances with gradient updates (Finn & Levine, 2017; Wang et al., 2020). This is particularly relevant in scenarios when adaptation of the model is needed for unseen data during inference. MAML is widely used to update INR weights (Dupont et al., 2022a; Jeong & Shin, 2022; Niemeyer et al., 2022; Bamford et al., 2023), however, its reliance on a test-time optimization step for each sample introduces computational overhead scaling with the number of test instances.

### 2.2 TIME SERIES IMPUTATION AND FORECASTING

RNNs are frequently used for time series forecasting, due of their ability to capture sequential dependencies (Chung et al., 2015; Hewamalage et al., 2021; Che et al., 2018; Guo et al., 2016). However, they assume fixed frequencies and struggle with long-term dependencies. To address these limitations, LSTM networks incorporate memory cells that retain relevant historical information while discarding irrelevant data (Hochreiter, 1997; Hua et al., 2019; Chen et al., 2022). Recent advancements have also embraced transformer-based architectures for time series modeling. Models such as SAITS (Du et al., 2023), PatchTST (Nie et al., 2023) and iTransformer (Liu et al., 2023) leverage attention and embedding strategies to capture both short- and long-term time dependencies within time series. Despite their strengths, transformers are inherently discrete and may fail to

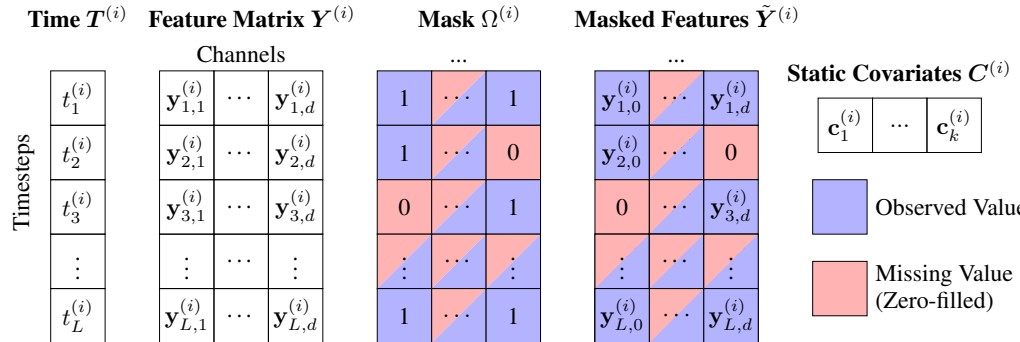

Figure 1: Visualization of temporal stamps $\boldsymbol{T}$, features $\boldsymbol{Y}$, mask $\Omega$, and static covariates $\boldsymbol{C}$. $\boldsymbol{T}$ and $\boldsymbol{Y}$ represent the input signal, $\Omega$ indicates missing values with binary entries, and $\boldsymbol{C}$ contains time-invariant covariates.

interpolate between time steps unless they are carefully redesigned for this task (Chen et al., 2024). Moreover, they may have trouble identifying and preserving key information when attending to large inputs (Wen et al., 2022). Likewise, conditional diffusion models like CSDI operate on fixed temporal grids and rely on architectural workarounds to manage irregular observations (Tashiro et al., 2021).

Recently, INRs have been used in continuous modeling of time series data for imputation and forecasting tasks (Naour et al., 2024; Fons et al., 2022; Cho et al., 2024), and for anomaly detection (Jeong & Shin, 2022). Fons et al. (2022) use a set-encoder approach to generate latent representations to parameterize INRs through hypernetworks for time series generation. Similarly, Bamford et al. (2023) adopt this approach for time series imputation, utilizing an auto-decoding strategy that requires back-propagation to learn these latent representations. Naour et al. (2024); Cho et al. (2024); Woo et al. (2023) use gradient-based meta-learning approaches to learn per instance modulations on INRs to perform imputation and forecasting on test data. Therefore, these methods encounter scalability challenges with an increasing number of test instances, since each requires per-instance optimization, and they may underperform in scenarios characterized by limited data availability.

## 3 TEMPORAL VARIATIONAL IMPLICIT NEURAL REPRESENTATIONS

In this section, we introduce **T**emporal **V**ariational **I**mplicit **N**eural **R**epresentations (TV-INRs). Our approach is motivated by representing time series as continuous functions using Implicit Neural Representations (INRs). Leveraging the amortized inference framework of Variational Autoencoders (Kingma, 2013; Rezende et al., 2014), TV-INRs learns distributions over INR parameters through encoder networks, eliminating per-sample optimization during inference while enabling efficient scaling to large datasets (Cremer et al., 2018; Hoffman et al., 2013; Mnih & Gregor, 2014). This approach maintains competitive performance for time series modeling tasks such as imputation and forecasting while facilitating personalized modeling through latent variables.

**Notation.** Let $[L] = \{1, \ldots, L\}$ denote the set of positive integers from 1 to $L$ and $d$ denote the total number of feature dimensions. We consider a dataset of $N$ samples $\{(\boldsymbol{T}^{(i)}, \boldsymbol{Y}^{(i)}, \boldsymbol{C}^{(i)})\}_{i=1}^N$, where each sample $i \in [N]$ as shown in Fig. 1 includes:

- **Temporal stamps**: A point cloud of $L_i$ temporal stamps (i.e. *temporal* coordinates), $\boldsymbol{T}^{(i)} = \{t_l^{(i)}\}_{l=1}^{L_i}$, with $t \in \mathbb{R}$.

- **Feature vectors**: Corresponding feature vectors $\boldsymbol{Y}^{(i)} = \{\mathbf{y}_l^{(i)}\}_{l=1}^{L_i}$, where $\mathbf{y}_l^{(i)} \in \mathbb{R}^{d_l^{(i)}}$ with $d_l^{(i)} \leq d$ representing the number of observed channels at index $l$. The set $\mathcal{A}^{(i)}$ identifies indexes $(l)$ where channels $(j)$ are absent in the original dataset.

- **Static covariates**: Static covariates $\boldsymbol{C}^{(i)} = \{\mathbf{c}^{(i)}\}$, where $\mathbf{c} \in \mathbb{R}^k$, which are constant for all stamps in the sample.

We denote the multichannel $i$-th time series as a tuple $\boldsymbol{X}^{(i)} = (\boldsymbol{T}^{(i)}, \boldsymbol{Y}^{(i)})$, consisting of $L_i$ (irregular) temporal stamps and their corresponding features. To effectively handle missing data, we distinguish between three sets of indices. The observed indices $\mathcal{O}^{(i)}$ represent available data points in our dataset, which we input to the model. The masked indices $\mathcal{M}^{(i)}$ correspond to entries we artificially mask during training to facilitate self-supervised learning and improve generalization to missing data

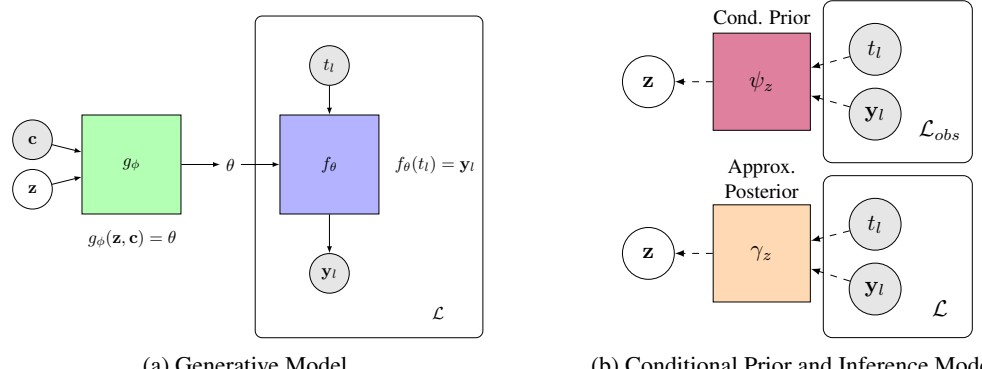

(a) Generative Model          (b) Conditional Prior and Inference Model

Figure 2: Graphical models for generative and inference tasks.

scenarios (Moreno-Muñoz et al., 2023). Finally, the absent indices $\mathcal{A}^{(i)}$ are inherent to the data and represent entries of missing channels due to partial observations or limitations in data collection which we exclude from the training process as they represent inherent data incompleteness rather than synthetic masks. We define a binary mask $\Omega^{(i)}$ to formalize this as:

$$\Omega_{l,k}^{(i)} = \begin{cases} 1 & \text{if } (l,k) \in \mathcal{O}^{(i)} \\ 0 & \text{if } (l,k) \in \mathcal{M}^{(i)} \\ 0 & \text{if } (l,k) \in \mathcal{A}^{(i)} \end{cases} \tag{1}$$

where $\mathcal{O}^{(i)}, \mathcal{M}^{(i)}, \mathcal{A}^{(i)} \subseteq [L_i] \times [d]$ with $\mathcal{O}^{(i)} \cap \mathcal{M}^{(i)} = \emptyset$. Finally, we denote by $\tau$ the percentage of observed indices in the available data, i.e., $\tau = \frac{|\mathcal{O}^{(i)}|}{|\mathcal{O}^{(i)} \cup \mathcal{M}^{(i)}|}$.

### 3.1 Model description

**Generative model.** To ease readability, we consider the model for a single sample and omit the use of the superscript $(i)$. TV-INRs is generative model for the feature set $\boldsymbol{Y}$ given timestamps $\boldsymbol{T}$. For now, we assume that $(\boldsymbol{T}, \boldsymbol{Y})$ is a timeseries with $L$ elements and $d$ channels without any absence, e.g. $\mathcal{A} = \emptyset$. The observed data $\boldsymbol{Y}_{\text{obs}}$ indexed by $\mathcal{O}^{(i)}$ and corresponding timestamps $\boldsymbol{T}_{\text{obs}}$ are given as context to the model, while $\boldsymbol{Y}_{\text{m}}$ indexed by $\mathcal{M}^{(i)}$ represents the masked values to predict at given timestamps $\boldsymbol{T}_{\text{m}}$. Together, they form the complete datasets: $\boldsymbol{Y} = \boldsymbol{Y}_{\text{m}} \cup \boldsymbol{Y}_{\text{obs}}$ and $\boldsymbol{T} = \boldsymbol{T}_{\text{m}} \cup \boldsymbol{T}_{\text{obs}}$ with the assumption of $\mathcal{A} = \emptyset$. The joint distribution can be written in a general form

$$p(\boldsymbol{Y}_{\text{m}}, \boldsymbol{Y}_{\text{obs}}, \mathbf{z} | \boldsymbol{Y}_{\text{obs}}, \boldsymbol{T}, \mathbf{c}) = p_{\boldsymbol{\psi_z}}(\mathbf{z} | \boldsymbol{Y}_{\text{obs}}, \boldsymbol{T}_{\text{obs}}) \prod_{l=1}^{L} p_{\boldsymbol{\theta}(\mathbf{z}, \mathbf{c})}(\mathbf{y}_l | t_l) \tag{2}$$

where $\mathbf{z}$ represents a latent variable and $\mathbf{c}$ denotes covariates. To generate such a signal, the process begins by sampling a continuous latent variable $\mathbf{z}$ from a conditional prior distribution, $p_{\boldsymbol{\psi_z}}(\mathbf{z} | \boldsymbol{Y}_{\text{obs}}, \boldsymbol{T}_{\text{obs}}) = \mathcal{N}(\mathbf{z} | f_{\boldsymbol{\psi_z}}(\boldsymbol{Y}_{\text{obs}}, \boldsymbol{T}_{\text{obs}}))$, which is parameterized by $\boldsymbol{\psi_z}$ using a Transformer encoder. The resulting vector $\mathbf{z}$, concatenated with random variable $\mathbf{c}$, acts as input to the *hyper-generator*. Here, the hypergenerator is an MLP-based hypernetwork $g_{\phi_k}(\mathbf{z}, \mathbf{c})$, with input $[\mathbf{z}, \mathbf{c}]$ that outputs a set of parameters $\boldsymbol{\theta}_k = g_{\phi_k}(\mathbf{z}, \mathbf{c})$; and, a data generator, $f_\theta$, parametrized by the output of the hypernetwork. Thus, both $\mathbf{z}$ and $\boldsymbol{\theta}$ encode the information shared among the stamps in the data (e.g., features) generation process as shown in Fig. 2a. Moreover, we refer to TV-INRs as C-TV-INRs when covariates are available and used.

**Inference model.** We approximate posterior distribution as $q_{\boldsymbol{\gamma_z}}(\mathbf{z} | \boldsymbol{Y}, \boldsymbol{T}) = \mathcal{N}(\mathbf{z} | f_{\boldsymbol{\gamma_z}}(\boldsymbol{Y}, \boldsymbol{T}))$, parameterized by $\boldsymbol{\gamma_z}$. It's important to note that this distribution is shared among the complete instance (e.g., time series signal), thus $\mathbf{z}$ contains global information as shown in Fig. 2b.

**Training.** We employ masked training by maximizing the evidence lower bound (ELBO) of the proposed model, which is given by

$$\mathcal{L}(\boldsymbol{T}, \boldsymbol{Y}, \boldsymbol{C}) = \mathbb{E}_{q_\gamma} \left[ \log p_{\boldsymbol{\theta}(\mathbf{z}, \mathbf{c})}[\boldsymbol{Y} \mid \boldsymbol{T}] \right] - D_{\text{KL}} \left( q_{\boldsymbol{\gamma_z}}(\mathbf{z} \mid \boldsymbol{Y}, \boldsymbol{T}) \| p_{\boldsymbol{\psi_z}}(\mathbf{z} | \boldsymbol{Y}_{\text{obs}}, \boldsymbol{T}_{\text{obs}}) \right) \tag{3}$$

where $p_{\boldsymbol{\psi_z}}$ and $q_{\boldsymbol{\gamma_z}}$ are Gaussian distributions, and we model $p_{\boldsymbol{\theta}(\mathbf{z}, \mathbf{c})}$ with a Laplace distribution as it demonstrates better performance in capturing high-frequency components.

## 3.2 IMPLEMENTATION DETAILS

We model the conditional prior and approximate posterior with Transformer encoders. To handle heterogeneity in the input data, we augment the input features by concatenating them with a binary mask, $(\Omega^{(i)} \in {0, 1}^{L_i \times d})$, which indicates observed entries across both temporal and feature dimensions.

**Input processing.** For each sample $i \in [N]$, we process the input tuple $(\boldsymbol{T}^{(i)}, \boldsymbol{Y}^{(i)}, \boldsymbol{C}^{(i)})$ to handle missing values. We construct the input representation using the binary mask $(\Omega^{(i)})$ as follows:

1. Fill masked values in $\boldsymbol{Y}^{(i)}$ with zeros:

$$\tilde{\boldsymbol{Y}}_{l,k}^{(i)} = \begin{cases} \boldsymbol{Y}_{l,k}^{(i)} & \text{if } (l,k) \in \mathcal{O}^{(i)} \\ 0 & \text{if } (l,k) \in \mathcal{U}^{(i)} \end{cases} \tag{4}$$

where $\tilde{\boldsymbol{Y}}^{(i)} \in \mathbb{R}^{L_i \times d}$, in case for the input of the posterior encoder we give full available data.

2. Concatenate the mask along the feature dimension and transform the processed features with a linear layer for spatial encoding, which captures relationships among different channels, yielding $\boldsymbol{E}_{\text{spatial}}^{(i)} = f_{\text{linear}}(\bar{\boldsymbol{Y}}^{(i)}) \in \mathbb{R}^{L_i \times d_{\text{model}}}$, where $\bar{\boldsymbol{Y}}^{(i)} = [\tilde{\boldsymbol{Y}}^{(i)}; \Omega^{(i)}] \in \mathbb{R}^{L_i \times 2d}$.

3. Expand temporal coordinates with channel indices $\mathbf{v}_d = [0, ..., d-1]$ and encode them with Fourier Features (FoF) (Dupont et al., 2021): $\boldsymbol{E}_{\text{temporal}}^{(i)} = \text{FoF}(\bar{\boldsymbol{T}}^{(i)}) \in \mathbb{R}^{L_i \times d_{\text{model}}}$, where $\bar{\boldsymbol{T}}^{(i)} = \boldsymbol{T}^{(i)} \otimes \mathbf{v}_d \in \mathbb{R}^{L_i \times d}$.

The final embedding $\boldsymbol{E}^{(i)} = \boldsymbol{E}_{\text{spatial}}^{(i)} + \boldsymbol{E}_{\text{temporal}}^{(i)}$ is element-wise summed and then fed into the encoder.

**Encoding.** The embedded input $\boldsymbol{E}^{(i)}$ is processed through a transformer encoder to model the conditional distributions $p_{\psi_{\mathbf{z}}}(\mathbf{z}|\boldsymbol{Y}_{\text{obs}}, \boldsymbol{T}_{\text{obs}})$ and $q_{\gamma_z}(\mathbf{z}|\boldsymbol{Y}, \boldsymbol{T})$. The encoder takes $\boldsymbol{E}^{(i)}$, transforms the input through self-attention, applies pooling (POOL) over temporal dimension, and a feed-forward network (FFN) generates parameters to model the latent features $\mathbf{z}$:

$$\mathbf{z} \sim \mathcal{N}(\boldsymbol{\mu}, \boldsymbol{\Sigma}) \text{ where } \boldsymbol{\mu}, \boldsymbol{\sigma} = \text{FFN}(\text{POOL}(\boldsymbol{H})), \text{ and } \boldsymbol{H} = \text{Transformer}(\boldsymbol{E}^{(i)}) \tag{5}$$

where $\boldsymbol{\Sigma} = \text{diag}(\boldsymbol{\sigma}^2)$. Here, we make sure masked values are not used during attention computation.

**Decoding.** The latent representation ($\mathbf{z}$) is combined with conditional variables to construct the decoder input through the following steps:

1. The conditional variables $\boldsymbol{C}^{(i)}$ are transformed by a feed-forward network into $\bar{\mathbf{c}} = \text{FFN}(\boldsymbol{C}^{(i)}) \in \mathbb{R}^{d_c}$, which is then concatenated with the latent representation to form the decoder input $\boldsymbol{h}_{\text{dec}} = [\mathbf{z}; \bar{\mathbf{c}}]$.

2. The resulting $\boldsymbol{h}_{\text{dec}}$ is passed through a hypernetwork $g_\phi$ to generate the parameters $\theta = g_\phi(\boldsymbol{h}_{\text{dec}})$ for the implicit neural representation (INR), $f_\theta$, which is continuous over $t$ (Sitzmann et al., 2020).

3. The INR, $f_\theta$, models the output feature values as $\hat{\boldsymbol{y}}_l \sim \text{Laplace}(\mu_l, b_l)$, where the distribution's parameters $(\mu_l, b_l) = f_\theta(\boldsymbol{e}_l)$ are the output of mapping the encoded time point $\boldsymbol{e}_l$.

## 4 EXPERIMENTS

**Baselines.** We thoroughly tested TV-INRs framework across imputation and forecasting tasks in full and limited data regimes with uni- and multi-variate datasets. We compare our model with TimeFlow (Naour et al., 2024), an INR-based time series model. It requires training separate models for different missingness ratios or horizon lengths, and performs gradient-based meta-learning during inference (details in App. A.8). We include two baselines specifically designed for time series imputation: SAITS (Du et al., 2023), which is based on self-attention, and CSDI (Tashiro et al., 2021), a conditional diffusion model that operates on a fixed temporal grid. For the forecasting task, we compare with DeepTime (Woo et al., 2023), which learns deep time-index models specifically designed for time series forecasting. Potential baselines HyperTime (Fons et al., 2022) and MADS (Bamford et al., 2023) were not available as open-source models, and were therefore not tested.

**Univariate datasets.** We conducted experiments on four univariate datasets (App. A.2 Table 4), and compared our approach to Timeflow (Naour et al., 2024), DeepTime (Woo et al., 2023), SAITS (Du et al., 2023), and CSDI Tashiro et al. (2021). Each dataset comprises one-dimensional signals originating from various locations or sources, and is available at the Monash Time Series Forecasting repository (Godahewa et al., 2021).

**Multivariate datasets.** While some datasets contain regular sampling (e.g., electricity), others are irregular, and have multiple sensors with unique temporal patterns. TV-INRs is the first temporal INR model to handle such multivariate signals, leading us to exclude Timeflow from these comparisons. We conducted experiments on two multivariate datasets, namely, HAR and The PhysioNet Challenge 2012 (P12), and compared our method with SAITS (Du et al., 2023) and CSDI (Tashiro et al., 2021). Additional details on the datasets, including missingness patterns, are provided in App. A.2.

Next, we describe the imputation and forecasting tasks. Let the $i$-th sample, $\boldsymbol{T}^{(i)} = \{t_j^{(i)}\}_{j=1}^{L_i}$, contain $L_i$ stamps. For both tasks, we compare predicted values against the ground truth for test data using Mean Squared Error (MSE) and Mean Absolute Error (MAE).

**Imputation task.** We partition the data based on an observed ratio $\tau$. Given the observed stamps $\boldsymbol{T}_{\text{obs}}^{(i)}$, the goal is to predict features at the unobserved stamps $\boldsymbol{T}_{\text{unobs}}^{(i)}$, where

$$\boldsymbol{T}^{(i)} = \boldsymbol{T}_{\text{obs}}^{(i)} \cup \boldsymbol{T}_{\text{unobs}}^{(i)}, \quad \boldsymbol{Y}^{(i)} = \boldsymbol{Y}_{\text{obs}}^{(i)} \cup \boldsymbol{Y}_{\text{unobs}}^{(i)}, \quad \hat{\boldsymbol{Y}}_{\text{unobs}} \sim p_{\boldsymbol{\theta}(\mathbf{z},\mathbf{c})}(\boldsymbol{Y}_{\text{unobs}} \mid \boldsymbol{T}_{\text{unobs}}). \tag{6}$$

The task's difficulty increases as $\tau$ decreases. For prediction, we use the conditional prior distribution $p_{\boldsymbol{\psi}_{\mathbf{z}}}(\mathbf{z}|\boldsymbol{Y}_{\text{obs}}, \boldsymbol{T}_{\text{obs}})$ and covariates $\mathbf{c}$ (if available).

**Forecasting task.** We partition data at a horizon $t_{\text{horizon}}$ into history and forecast sets. Given the observed historical data $\boldsymbol{Y}_{\text{hist}}^{(i)}$, our task is to predict $\boldsymbol{Y}_{\text{forecast}}^{(i)}$. We use our conditional prior $p_{\boldsymbol{\psi}_{\mathbf{z}}}(\mathbf{z}|\boldsymbol{Y}_{\text{hist}}, \boldsymbol{T}_{\text{hist}})$ and covariates $\mathbf{c}$ (if available) to generate predictions:

$$\boldsymbol{T}_{\text{hist}}^{(i)} = \{t_j^{(i)} \in \boldsymbol{T}^{(i)} \mid t_j^{(i)} \leq t_{\text{horizon}}\}, \boldsymbol{T}_{\text{forecast}}^{(i)} = \{t_j^{(i)} \in \boldsymbol{T}^{(i)} \mid t_j^{(i)} > t_{\text{horizon}}\} \tag{7}$$

$$\hat{\boldsymbol{Y}}_{\text{forecast}} \sim p_{\boldsymbol{\theta}(\mathbf{z},\mathbf{c})}(\boldsymbol{Y}_{\text{forecast}} \mid \boldsymbol{T}_{\text{forecast}}). \tag{8}$$

## 4.1 RESULTS

In Sections 4.1.1 and 4.1.2, we explore TV-INRs performance in imputation and forecasting on univariate datasets in comparison with the baseline models Timeflow (Naour et al., 2024), SAITS (Du et al., 2023), CSDI (Tashiro et al., 2021) and DeepTime (Woo et al., 2023). We comment on the training efficiency in Sections 4.1.3 and App. A.10. In Section 4.1.4, we report TV-INRs performance on multivariate datasets including the conditional version of our model, C-TV-INRs, compared with SAITS (Du et al., 2023) and CSDI (Tashiro et al., 2021). Statistical significance ($p < 0.05$) was assessed using independent t-tests performed on results from non-overlapping test windows and three different seeds of model training. Ablation studies on the number of Fourier Features and our INR-based decoder are in App. A.12 and A.13, respectively. The code will be accessible in our repository.

### 4.1.1 IMPUTATION ON UNIVARIATE DATASETS

For imputation, we compared TV-INRs against the selected baselines across varying signal lengths $L$. We used $L = 2000$ (2K) time points to match published baseline experiments, and $L = 200$ time points to evaluate performance in lower-data regimes. We define the rate of observed data points during testing as $\tau_{Test}$. The **low-data regime** is characterized by conditions of data scarcity, which includes all scenarios with a limited training set ($L = 200$) and sparse test-time observations $\tau_{Test} \in \{0.5, 0.3, 0.05\}$) as well as the experiments with a larger training set but very sparse test-time observations ($L = 2000$, $\tau_{Test} = 0.05$). In contrast, **the high-data regime** represents scenarios with a relative abundance of data, specifically when a larger training set is available ($L = 2000$) and the observation rates at test time are higher ($\tau_{Test} \in \{0.5, 0.3\}$) or when $L = 10000$ and $\tau_{Test} \in \{0.5, 0.3, 0.05\}$. To improve robustness under low observation rates, we sample the observed fraction at random during training, e.g. $\tau_{Train} \sim S = \{0.05, 0.30, 0.50, 0.75, 0.90, 1.0\}$. TimeFlow requires separate training for each $\tau_{Test}$ value, while SAITS fixes $\tau_{Train} = 0.80$ and CSDI uses a uniform distribution $\tau_{Train} \sim \mathcal{U}(0, 1)$.

The results in Table 1 demonstrate the advantages of our approach over gradient-based meta-learning, particularly in low-data regimes. With shorter signals ($L = 200$) and lower observation percentages $\tau_{Test}$, TV-INRs consistently performs on par or better than all baselines, achieving up to $88\%$ improvement in MSE scores. In Solar-10 at ($L = 200$) specifically, TV-INRs achieves substantially lower error rates, with a MSE of 0.0383 compared to TimeFlow's 0.3304, SAITS' 0.0660 and CSDI's 1.010 at $\tau_{Test} = 0.50$. At the highest missingness setting, $\tau_{Test} = 0.05$, TV-INRs also performs best on average, though it is only comparable to TimeFlow on the Solar-10 dataset. As Solar-10 has significantly longer time series ($L = 10K$) and thus a larger number of training observations, results indicate that TV-INRs excels primarily in low-data regimes.

Table 1: **Univariate imputation results** with signal lengths $L$, training/testing observation rates $\tau_{\text{train,test}}$, and MSE/MAE evaluated on unobserved indices from non-overlapping test signals. Bold values indicate significantly better results, while underlined values denote results that are comparable.

| Model | $L$ | $\tau_{\text{Train}}$ | $\tau_{\text{Test}}$ | Electricity MSE | Electricity MAE | Traffic MSE | Traffic MAE | $L$ | Solar-10 MSE | Solar-10 MAE |
|---|---|---|---|---|---|---|---|---|---|---|
| SAITS | 2K | 0.80 | 0.50 | 0.569 ± 0.048 | 0.542 ± 0.022 | 0.251 ± 0.028 | 0.246 ± 0.015 | | 1.086 ± 0.005 | 0.648 ± 0.022 |
| | | | 0.30 | 0.793 ± 0.055 | 0.654 ± 0.023 | 0.337 ± 0.033 | 0.306 ± 0.015 | 10K | 1.087 ± 0.009 | 0.651 ± 0.024 |
| | | | 0.05 | 1.318 ± 0.051 | 0.902 ± 0.025 | 0.824 ± 0.040 | 0.619 ± 0.014 | | 1.126 ± 0.061 | 0.676 ± 0.062 |
| CSDI | 2K | $\sim \mathcal{U}$ | 0.50 | 2.070 ± 0.194 | 1.033 ± 0.023 | 1.150 ± 0.029 | 0.773 ± 0.144 | | 1.275 ± 0.382 | 0.699 ± 0.781 |
| | | | 0.30 | 2.287 ± 0.157 | 1.045 ± 0.012 | 1.146 ± 0.103 | 0.773 ± 0.165 | 10K | 1.285 ± 0.191 | 0.703 ± 0.749 |
| | | | 0.05 | 1.742 ± 0.265 | 1.050 ± 0.013 | 1.139 ± 0.111 | 0.773 ± 0.171 | | 1.279 ± 0.020 | 0.700 ± 0.737 |
| TimeFlow | 2K | 0.50 | 0.50 | **0.131 ± 0.011** | **0.252 ± 0.010** | **0.346 ± 0.036** | **0.369 ± 0.017** | | **0.710 ± 0.040** | 0.617 ± 0.056 |
| | | 0.30 | 0.30 | **0.166 ± 0.012** | **0.288 ± 0.011** | **0.390 ± 0.042** | 0.388 ± 0.018 | 10K | **0.812 ± 0.128** | 0.658 ± 0.121 |
| | | 0.05 | 0.05 | 0.378 ± 0.034 | 0.458 ± 0.025 | 0.590 ± 0.048 | 0.496 ± 0.020 | | 0.833 ± 0.010 | 0.663 ± 0.096 |
| TV-INRs | 2K | $\sim \mathcal{S}$ | 0.50 | 0.249 ± 0.019 | 0.331 ± 0.012 | 0.546 ± 0.022 | 0.401 ± 0.015 | | 0.955 ± 0.059 | 0.645 ± 0.038 |
| | | | 0.30 | 0.250 ± 0.017 | 0.332 ± 0.012 | 0.551 ± 0.029 | 0.403 ± 0.017 | 10K | 0.954 ± 0.074 | 0.646 ± 0.050 |
| | | | 0.05 | **0.289 ± 0.019** | **0.360 ± 0.015** | 0.570 ± 0.019 | **0.415 ± 0.013** | | 1.104 ± 0.265 | 0.688 ± 0.132 |
| SAITS | 200 | 0.80 | 0.50 | 0.124 ± 0.014 | 0.223 ± 0.010 | 0.230 ± 0.015 | 0.245 ± 0.008 | | 0.066 ± 0.035 | 0.140 ± 0.021 |
| | | | 0.30 | 0.231 ± 0.025 | 0.317 ± 0.017 | 0.345 ± 0.019 | 0.320 ± 0.009 | 200 | 0.099 ± 0.060 | 0.168 ± 0.030 |
| | | | 0.05 | 0.937 ± 0.040 | 0.743 ± 0.018 | 0.904 ± 0.020 | 0.641 ± 0.016 | | 0.564 ± 0.107 | 0.502 ± 0.037 |
| CSDI | 200 | $\sim \mathcal{U}$ | 0.50 | 1.380 ± 0.216 | 0.944 ± 0.035 | 1.169 ± 0.204 | 0.787 ± 0.187 | | 1.010 ± 0.261 | 0.602 ± 0.122 |
| | | | 0.30 | 1.399 ± 0.144 | 0.945 ± 0.021 | 1.167 ± 0.183 | 0.789 ± 0.194 | 200 | 1.052 ± 0.209 | 0.625 ± 0.109 |
| | | | 0.05 | 1.226 ± 0.065 | 0.911 ± 0.011 | 1.158 ± 0.200 | 0.795 ± 0.194 | | 1.196 ± 0.716 | 0.700 ± 0.124 |
| TimeFlow | 200 | 0.50 | 0.50 | 0.163 ± 0.009 | 0.240 ± 0.007 | 0.233 ± 0.009 | 0.230 ± 0.006 | | 0.330 ± 0.046 | 0.223 ± 0.032 |
| | | 0.30 | 0.30 | 0.331 ± 0.014 | 0.396 ± 0.010 | 0.419 ± 0.015 | 0.370 ± 0.009 | 200 | 0.518 ± 0.057 | 0.331 ± 0.038 |
| | | 0.05 | 0.05 | 0.963 ± 0.019 | 0.811 ± 0.011 | 1.303 ± 0.103 | 0.830 ± 0.028 | | 0.877 ± 0.077 | 0.707 ± 0.098 |
| TV-INRs | 200 | $\sim \mathcal{S}$ | 0.50 | 0.113 ± 0.018 | **0.212 ± 0.015** | 0.188 ± 0.041 | 0.212 ± 0.027 | | **0.038 ± 0.031** | **0.089 ± 0.035** |
| | | | 0.30 | **0.135 ± 0.027** | **0.232 ± 0.021** | **0.214 ± 0.042** | **0.228 ± 0.028** | 200 | **0.051 ± 0.051** | **0.098 ± 0.042** |
| | | | 0.05 | **0.318 ± 0.063** | **0.368 ± 0.041** | **0.453 ± 0.074** | **0.368 ± 0.042** | | **0.244 ± 0.226** | **0.234 ± 0.099** |

For longer signal lengths ($L = 2K, 10K$), TimeFlow shows stronger performance on the Electricity and Traffic datasets at higher $\tau_{\text{Test}}$ values. Overall, TV-INRs *maintains competitive performance across all scenarios while offering two crucial advantages: it provides a unified model that handles all cases without requiring per-case training, and enables efficient inference through gradient-free meta-learning that requires only a forward pass.* These results highlight how our variational framework effectively balances performance with practical efficiency, and excels in scenarios where data availability is limited. In App. B.1, Figures 4-5 show sample outputs generated by TV-INRs.

### 4.1.2 FORECASTING ON UNIVARIATE DATASETS

For forecasting, we compare TV-INRs with TimeFlow and DeepTime using the same experimental settings as in their original publications. The historical length $H$ is set to the first 512 elements, and forecasting performance is evaluated over forecasting lengths $F$ of 96, 192, 336, and 720. TV-INRs is trained by sampling forecasting lengths $F_{\text{Train}} \in \mathcal{F} = \{96, 192, 336, 720\}$. Since $H$ is fixed, the binary mask has the same number of observed indices; however, the total length of the mask is adapted to different lengths of $F$. As shown in Table 2, both TimeFlow and DeepTime require separate training for each forecasting length, while our approach uses a single model for all horizons. For TV-INRs and TimeFlow, there is a dramatic increase in MSE for long-range forecasting ($F = 720$) in the Electricity dataset, reaching $\approx 9.5$ and $\approx 9.4$ respectively, while maintaining relatively moderate MAE ($\approx 0.53$), which strongly indicates the presence of significant outlier errors in the predictions. DeepTime shows even higher errors in this scenario (MSE = 10.18). For shorter forecasting horizons ($F = \{96, 192\}$), our method demonstrates competitive or superior performance, notably achieving a MSE of 0.3359 versus TimeFlow's 0.4250 and DeepTime's 0.4359 for $F = 96$ in the Electricity dataset. Our approach significantly outperforms DeepTime on the Solar-H dataset, with MSE of 0.3456 versus 0.6410 at $F = 96$. TimeFlow achieves lower errors in specific scenarios (Traffic at $F = 96$, Solar-H at $F = \{336, 720\}$), but requires separate training per horizon and gradient-based meta-learning for each test sample. Similarly, DeepTime needs individual models for each forecast length. Our approach's key advantage is *handling multiple forecasting horizons with a single trained model while maintaining competitive performance.* Sample outputs are shown in App. B.1 (Fig.6).

Table 2: **Univariate forecasting results** with history length $H$, training/testing forecasting lengths $F_{\text{train,test}}$, and MSE/MAE evaluated for forecasting. Bold values indicate significantly better results, while underlined values denote results that are comparable.

| Model | H | $F_{\text{train}}$ | $F_{\text{test}}$ | Electricity | | Traffic | | Solar-H | |
|---|---|---|---|---|---|---|---|---|---|
| | | | | MSE | MAE | MSE | MAE | MSE | MAE |
| DeepTime | 512 | 96 | 96 | 0.436 ± 0.020 | 0.503 ± 0.016 | 0.419 ± 0.103 | 0.411 ± 0.047 | 0.641 ± 0.183 | 0.651 ± 0.089 |
| | | 192 | 192 | 0.551 ± 0.157 | 0.525 ± 0.055 | 0.382 ± 0.056 | 0.372 ± 0.027 | 0.432 ± 0.121 | 0.514 ± 0.081 |
| | | 336 | 336 | 0.793 ± 0.046 | 0.689 ± 0.037 | 0.446 ± 0.107 | 0.397 ± 0.058 | 0.821 ± 0.013 | 0.804 ± 0.002 |
| | | 720 | 720 | 10.178 ± 0.218 | 0.970 ± 0.178 | 0.485 ± 0.059 | 0.406 ± 0.014 | 0.793 ± 0.041 | 0.741 ± 0.001 |
| TimeFlow | 512 | 96 | 96 | 0.425 ± 0.057 | 0.318 ± 0.050 | **0.289 ± 0.113** | 0.281 ± 0.064 | 0.503 ± 0.424 | 0.336 ± 0.142 |
| | | 192 | 192 | 0.498 ± 0.078 | **0.362 ± 0.060** | 0.324 ± 0.076 | 0.298 ± 0.050 | 0.476 ± 0.191 | 0.352 ± 0.077 |
| | | 336 | 336 | 1.347 ± 0.210 | **0.389 ± 0.065** | 0.407 ± 0.122 | 0.329 ± 0.057 | **0.364 ± 0.106** | **0.301 ± 0.055** |
| | | 720 | 720 | 9.422 ± 0.217 | 0.525 ± 0.150 | 0.413 ± 0.050 | 0.327 ± 0.020 | **0.353 ± 0.092** | **0.325 ± 0.032** |
| TV-INRs | 512 | $\sim \mathcal{F}$ | 96 | **0.336 ± 0.068** | 0.296 ± 0.040 | 0.383 ± 0.143 | 0.305 ± 0.082 | **0.346 ± 0.303** | 0.325 ± 0.123 |
| | | | 192 | 0.446 ± 0.107 | 0.415 ± 0.036 | 0.377 ± 0.094 | 0.294 ± 0.056 | 0.469 ± 0.125 | 0.389 ± 0.031 |
| | | | 336 | **0.544 ± 0.216** | 0.442 ± 0.040 | 0.373 ± 0.073 | **0.292 ± 0.049** | 0.451 ± 0.140 | 0.383 ± 0.039 |
| | | | 720 | 9.515 ± 0.218 | 0.535 ± 0.162 | 0.448 ± 0.088 | 0.313 ± 0.043 | 0.509 ± 0.194 | 0.404 ± 0.061 |

### 4.1.3 EXPLANATION OVER GENERALIZATION CLAIMS

We assess model generalization by its robust performance across a range of distinct tasks, each applied to $N$ unique time series. For imputation, these tasks are defined by varying the observation rate $\tau$, challenging the model under different levels of data scarcity. For forecasting, we measure generalization by the model's ability to maintain accuracy over increasingly long forecasting windows, $\mathcal{F} \in \{96, 192, 336, 720\}$. TV-INRs uses a unified model capable of imputation with different observed ratios and forecasting across all horizon lengths, which significantly reduces or eliminates the need for additional fine-tuning or multiple-model optimizations, enhancing its overall efficiency. To illustrate this, we show that TimeFlow has to be trained per scenario, e.g. different observed ratios and horizon lengths, in Table 19 in App.A.6. We report the training times for TV-INRs and TimeFlow across all experiments in App. A.10. Our findings indicate that TV-INRs achieves notable improvements in cumulative training efficiency: it requires between 2.41× to 3.70× less training time than TimeFlow for forecasting tasks, and between 1.30× to 2.81× less training time for imputation tasks. These results are shown in App. A.10 - Table 20, and demonstrate that TV-INRs offers substantial advantages in computational efficiency and generalization by handling multiple tasks with a single training. We also provide the memory and time complexity analysis of TV-INR in App. A.9.

### 4.1.4 IMPUTATION ON MULTIVARIATE DATASETS

**In the HAR dataset,** motion data from a single smartphone presents simultaneous missing values across all channels at specific timestamps due to device failures. Formally, given $\boldsymbol{X}^{(i)} = \boldsymbol{X}_{\text{obs}}^{(i)} \cup \boldsymbol{X}_{\text{unobs}}^{(i)}$, where $\boldsymbol{X}_{\text{unobs}}^{(i)} = \boldsymbol{X}_l^{(i)} : l \in \mathcal{U}^{(i)}$, any missing timestamp $l \in (\mathcal{U}^{(i)})$ affects all $d$ channels.

**For the P12 dataset,** we evaluate TV-INRs on patient-specific time series imputation from eight measurements (urine output, SysABP, DiasABP, MAP, HR, NISysABP, NIDiasABP, NIMAP) and four covariates (gender, age, height, weight). The dataset has irregular missingness across timestamps and channels, which makes the imputation task more challenging (details in App. A.2).

- **Conditional vs. unconditional.** We test C-TV-INRs conditional formulation (Equation 2) on HAR by incorporating activity labels alongside latent codes, and on P12 by including patient covariates. On HAR, Table 3 shows C-TV-INRs significantly outperforms TV-INRs at higher missingness rates ($\tau_{\text{Test}} = 0.05$). For P12, both variants perform comparably at higher observation rates ($\tau_{\text{Test}} = 0.50, 0.30$). But at extreme sparsity ($\tau_{\text{Test}} = 0.10$), C-TV-INRs significantly outperforms with MSE=0.9627 versus SAITS's 0.9704, CSDI's 1.024, and TV-INRs's 0.9795, with the lowest MAE (0.7326). This confirms conditional models' advantage with sparse time series data. Overall, both the conditional and non-conditional versions of TV-INRs outperform baselines for multivariate imputation.

- **Downstream classification.** To assess the impact of imputation on classification, we trained an XGBoost classifier (Chen & Guestrin, 2016) on HAR data, testing across varying observation ratios by removing random timepoints and imputing using our methods, baselines, and mean imputation. Fig. 3 shows both TV-INRs variants substantially outperforming baselines, with the conditional model showing increasing advantage as missingness grows, demonstrating the value of covariates for individualized predictions. Complete AUC-ROC values are in Table 11.

Table 3: **Multivariate imputation results** with signal lengths $L$, training/testing observation rates $\tau_{\text{train,test}}$, and MSE/MAE evaluated on unobserved indices from non-overlapping test signals. Bold values indicate significantly better results, while underlined values denote results that are comparable.

| Model | | | HAR (L=128) | | | | P12 (L=48) | |
| --- | --- | --- | --- | --- | --- | --- | --- | --- |
| | $\tau_{\text{Train}}$ | $\tau_{\text{Test}}$ | MSE | MAE | $\tau_{\text{Train}}$ | $\tau_{\text{Test}}$ | MSE | MAE |
| SAITS | 0.80 | 0.50 | $0.998 \pm 0.003$ | $0.793 \pm 0.006$ | 0.80 | 0.50 | $0.985 \pm 0.128$ | $0.746 \pm 0.070$ |
| | | 0.30 | $1.001 \pm 0.004$ | $0.793 \pm 0.007$ | | 0.30 | $0.998 \pm 0.092$ | $0.760 \pm 0.067$ |
| | | 0.05 | $1.004 \pm 0.001$ | $0.793 \pm 0.007$ | | 0.10 | $0.970 \pm 0.048$ | $0.746 \pm 0.052$ |
| CSDI | $\sim \mathcal{U}$ | 0.50 | $1.083 \pm 0.062$ | $0.821 \pm 0.067$ | $\sim \mathcal{U}$ | 0.50 | $0.861 \pm 0.174$ | $0.691 \pm 0.070$ |
| | | 0.30 | $1.084 \pm 0.060$ | $0.823 \pm 0.063$ | | 0.30 | $0.930 \pm 0.146$ | $0.724 \pm 0.067$ |
| | | 0.05 | $1.090 \pm 0.015$ | $0.826 \pm 0.054$ | | 0.10 | $1.024 \pm 0.093$ | $0.765 \pm 0.057$ |
| TV-INRs | $\sim \mathcal{S}$ | 0.50 | $\underline{0.382 \pm 0.067}$ | $\underline{0.414 \pm 0.041}$ | $\sim \mathcal{S}$ | 0.50 | $\underline{0.822 \pm 0.171}$ | $\underline{0.660 \pm 0.074}$ |
| | | 0.30 | $\underline{0.533 \pm 0.050}$ | $\underline{0.505 \pm 0.031}$ | | 0.30 | $\underline{0.892 \pm 0.146}$ | $\underline{0.692 \pm 0.071}$ |
| | | 0.05 | $0.995 \pm 0.070$ | $0.722 \pm 0.034$ | | 0.10 | $0.980 \pm 0.118$ | $0.739 \pm 0.058$ |
| C-TV-INRs | $\sim \mathcal{S}$ | 0.50 | $\underline{0.379 \pm 0.065}$ | $\underline{0.412 \pm 0.041}$ | $\sim \mathcal{S}$ | 0.50 | $\underline{0.824 \pm 0.175}$ | $\underline{0.662 \pm 0.076}$ |
| | | 0.30 | $\underline{0.523 \pm 0.047}$ | $\underline{0.502 \pm 0.029}$ | | 0.30 | $\underline{0.883 \pm 0.141}$ | $\underline{0.690 \pm 0.073}$ |
| | | 0.05 | $\mathbf{0.976 \pm 0.058}$ | $\mathbf{0.708 \pm 0.022}$ | | 0.10 | $\mathbf{0.963 \pm 0.099}$ | $\mathbf{0.733 \pm 0.052}$ |

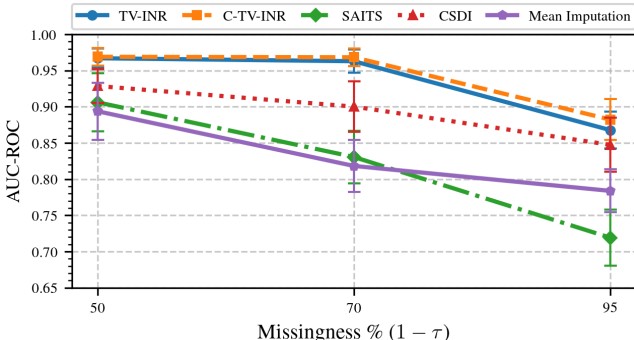

Figure 3: Classification performance (AUC-ROC) at various missingness levels; a higher value indicates better performance.

## 5 CONCLUSION

We have introduced TV-INRs, demonstrating its effectiveness in imputation and forecasting across various time series domains and data conditions. Our results highlight superior performance in low-data regimes and robust handling of varying observation patterns. Furthermore, the amortization of INR weights in our probabilistic setting enables adaptation to unseen data without fine-tuning or per-sample optimization, a key advantage over traditional hypernetwork-based methods that rely on meta-learning. We have also illustrated the potential of TV-INRs for downstream tasks with improved classification on HAR data. While baseline methods TimeFlow and DeepTime showed stronger performance in specific scenarios, TV-INRs frequently produced comparable or superior results while offering substantial practical benefits: unified model training across multiple tasks, individualization without meta-learning, and significantly improved cumulative training and inference efficiency. The ability to handle multiple forecasting horizons with a single model represents a considerable advantage in real-world applications where computational resources may be limited.

To further enhance our model, future directions may include reducing hypernetwork complexity with transformer-based architectures (Chen & Wang, 2022), or modeling per-sample positional embeddings rather than weights directly (Park et al., 2024). The variational framework could also be extended to incorporate additional forms of domain knowledge. These improvements could strengthen its potential, particularly in healthcare domains such as personalized medicine and patient monitoring, where efficiency and the ability to model highly sparse data are especially critical.

## 6 REPRODUCIBILITY STATEMENT

To ensure the reproducibility of our results, we have included our complete source code as supplementary material. Our code submission contains the model implementation, training scripts for experiments, and instructions for setting up the required environment. Furthermore, a detailed description of all experimental settings, including dataset preprocessing steps (App. A.3,A.5) and the final hyperparameter configurations (App. A.6) are provided.

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

# A    APPENDIX A

## A.1    THE USE OF LARGE LANGUAGE MODELS (LLMs)

The authors used a large language model (LLM) as a general-purpose writing assistant during the preparation of this paper. Its application was exclusively for grammar checking. The LLM played no role in the research ideation, methodology, or the generation of the core manuscript content, which remains the sole contribution of the authors.

## A.2    DATASETS

Table 4: **Dataset Descriptions.** #Series denotes the number of distinct timeseries signals with corresponding lenghts and covariates if available.

| Dataset | Domain | Freq. | #Dims | #Series | Length | Cov. |
|---|---|---|---|---|---|---|
| Electricity | $\mathbb{R}_0^+$ | Hourly | 1 | 321 | 26304 | ✗ |
| Traffic | [0,1] | Hourly | 1 | 862 | 17544 | ✗ |
| Solar-10 | $\mathbb{R}_0^+$ | 10 Mins | 1 | 137 | 52560 | ✗ |
| Solar-H | $\mathbb{R}_0^+$ | Hourly | 1 | 137 | 8760 | ✗ |
| HAR | $\mathbb{R}$ | 50Hz | 3 | 30 | 43940 | ✓ |
| P12 | $\mathbb{R}_0^+$ | Hourly | 8 | 3938 | 48 | ✓ |

In this section, we provide more details about the datasets we have used. We start with the list of uni-variate datasets:

**Electricity Dataset** records hourly electricity consumption from 321 customers in Portugal for the period 2012 to 2014, displaying both daily and weekly seasonality.

**Traffic Dataset** includes hourly road occupancy rates from 862 locations in San Francisco during 2015 and 2016, and exhibits similar daily and weekly seasonal patterns.

**Solar Dataset** The Solar-10 dataset comprises measurements of solar power production from 137 photovoltaic plants in Alabama, captured every 10 minutes in 2006. Additionally, there is an hourly version of this dataset, known as Solar-Hourly.

For some datasets, the feature vectors $Y^{(i)} = \{y_l^{(i)}\}_{l=1}^{L_i}$ expand from univariate ($d = 1$) to multivariate ($d > 1$), with each dimension representing a unique sensor used to collect observations $\{y_l^{(i)}\} \in \mathbb{R}^d$. For these purposes, we experiment with two multi-variate datasets, namely:

**HAR Dataset.** Here, we experiment with the Human Activity Recognition (HAR) dataset from the UC Irvine ML Repository, which is dense with regular time points at $2.56$ second intervals, enabling quantitative imputation assessment through random removal. It contains 10,299 samples of accelerometer measurements across x, y, and z axes.

**P12 Dataset.** The PhysioNet Challenge 2012 (P12) dataset contains ICU stay measurements including sensor readings and lab results. After outlier removal, it comprises 11,817 visits across 37 channels with maximum 215 time points over 48 hours. We use eight measurements urine output, systolic arterial blood pressure (SysABP), diastolic arterial blood pressure (DiasABP), mean arterial pressure (MAP), heart rate (HR), and their non-invasive counterparts (NISysABP, NIDiasABP, NIMAP). We also incorporate patient-specific covariates including gender, age, height, and weight. Conditional TV-INRs use covariates Unlike HAR, P12 is highly sparse ($X_{\text{obs}}^{(i)}$ is 15.68% of $X$ on average) with irregularity across times and sensors, where $T^{(i)}$ may be unique for each time series $i$.

**Missingness Patterns of the Datasets.** To ensure a comprehensive evaluation, our experiments address diverse data missingness patterns, including both random and non-random scenarios. For Missing Completely at Random (MCAR) patterns, we adhere to standard literature practices by introducing artificial missingness (Little & Rubin, 2019) during training across the Electricity, Traffic, and Solar datasets. This methodology aligns with the protocols used by the baseline models we compare against. Furthermore, we assess performance on Missing Not at Random (MNAR) patterns, which are prevalent in real-world applications. Our analysis includes the P12 dataset, which exhibits MNAR characteristics where clinical data is informatively missing; here, we evaluate imputation quality indirectly via a downstream classification task. To create a controlled non-random evaluation, we also synthetically modified the fully-observed HAR dataset by dropping entire channels at random timestamps to mimic sensor failures, a scenario where the missingness mechanism depends on unobserved factors.

## A.3 DATA-PREPROCESSING

We apply channel-wise standardization to each time series. For each channel $d$ in a time series with length $L$, we compute the channel-wise mean $\mu_d$, standard deviation $\sigma_c$, and normalize signal $\hat{x}_{l,d}^{(i)}$ as follows:

$$\hat{x}_{l,d}^{(i)} = \frac{x_{l,d}^{(i)} - \mu_d^{(i)}}{\sigma_d^{(i)}} \tag{9}$$

where $x_{l,d}^{(i)}$ represents the value of channel $d$ at time $l$ for sample $i$.

## A.4 ANALYSIS FOR STATISTICAL DIFFERENCES

To compare the performance of TV-INRs and baseline models, we conducted a systematic statistical analysis using Welch's t-test which accounts for potentially unequal variances between the two models. For each configuration defined by sequence length $L$ and sampling ratio $\tau$, we evaluated both mean squared error (MSE) and mean absolute error (MAE). The statistical significance was assessed at $\alpha = 0.05$.

In classification experiments, the HAR dataset was normalized independently per channel but not per individual, ensuring consistency across subjects and allowing XGBoost to learn global patterns. This differs from the normalization procedure used for TV-INRs, which normalized data at both the channel and individual level in order to model data on a per-user basis. When mentioned, we computed the relative performance difference as $\Delta = (\mu_{\text{TimeFlow}} - \mu_{\text{TV-INRs}})/\mu_{\text{TimeFlow}} \times 100\%$.

## A.5 TRAINING, VALIDATION, AND TEST SPLITS FOR ALL EXPERIMENTS

Here, we give information about all datasplits for all experiments in Tables 5, 6, 7. For univariate datasets, test windows are extracted sequentially from the end of each time series. Moreover, training data precedes validation data.

Table 5: Dataset splitting details for univariate imputation experiments. Training and validation sets has 5:1 ratio.

| Dataset | Series Count | Window Length (L) | Test Windows (NO & FE)[1] | Training/Val. Stride |
|---|---|---|---|---|
| Electricity | 321 | 200 | 50 | 50 |
| | | 2000 | 5 | 500 |
| Traffic | 862 | 200 | 20 | 50 |
| | | 2000 | 2 | 500 |
| Solar-10 | 137 | 200 | 100 | 50 |
| | | 10000 | 2 | 250 |

Table 6: Dataset splitting details for univariate forecasting experiments. Training and validation sets has 5:1 ratio. Training and validation series are constructed with using offsetting from the available data points.

| Dataset | Series Count | History (H) | Forecast (F) | Window Length (L) | Test Windows (NO & FE)[2] | Training/Val. Offset |
|---|---|---|---|---|---|---|
| Electricity | 321 | 512 | [96,192,336,720] | 1232 | 7 | ✓ |
| Traffic | 862 | 512 | [96,192,336,720] | 1232 | 7 | ✓ |
| Solar-H | 137 | 512 | [96,192,336,720] | 1232 | 3 | ✓ |

---

[1]NO: Non-overlapping, FE: From end of the series

Table 7: Dataset splitting details for HAR imputation experiments. The dataset is split by users, with 24 users for training and 6 users for testing. From the training users, we further split into training and validation sets using a 4:1 ratio of users.

| Dataset | Series Count | Window Length (L) | #Classes | #Train Users | #Test Users |
|---|---|---|---|---|---|
| HAR | 30 | 128 | 6 | 24 | 6 |
| P12 | 11817 | 48 | NA | 9454 | 2363 |

## A.6 HYPERPARAMETERS FOR ALL EXPERIMENTS

Hyperparameters for all TV-INR experiments on an NVIDIA V100 GPU can be seen in Tables 8-9. In case of HAR dataset, C-TV-INRs extra parameters of feed forward encoder of covariates with layers $[8, 8]$ and dim_c $= 4$. The details of the hyperparameter grid search space are provided in Table 10.

Table 8: Hyperparameter details of TV-INRs for imputation task.

|  |  | ELECTRICITY | | TRAFFIC | | SOLAR-10 | | HAR |
|---|---|---|---|---|---|---|---|---|
|  | L | 200 | 2000 | 200 | 2000 | 200 | 10000 | 128 |
|  | dim_z | 32 | 64 | 32 | 64 | 32 | 64 | 32 |
|  | epochs | 2000 | 4000 | 2000 | 4000 | 2000 | 4000 | 3000 |
|  | bs | 256 | 64 | 256 | 64 | 256 | 32 | 128 |
|  | lr | 1e-4 | 1e-4 | 1e-4 | 1e-4 | 1e-4 | 1e-4 | 1e-4 |
| Transformer Enc. | $d_{model}$ | 128 | 128 | 128 | 128 | 128 | 128 | 128 |
|  | #heads | 2 | 4 | 2 | 4 | 2 | 4 | 4 |
|  | #layers | 2 | 2 | 2 | 2 | 2 | 2 | 4 |
| Hypernetwork | layers | [128,256] | | | | | | |
| Generator | layers | [64,64,64] | [64,64,64,64] | [64,64,64] | [64,64,64,64] | [64,64,64] | [64,64,64,64] | [64,64,64,64] |
| RFF |  | $m = 256, \sigma = 2$ | | | | | | |

Table 9: Hyperparameter details of TV-INRs for forecasting task.

|  |  | ELECTRICITY | TRAFFIC | SOLAR-H |
|---|---|---|---|---|
|  | dim_z | 32 | 64 | 32 |
|  | max epochs | 2000 | 4000 | 2000 |
|  | bs | 256 | 64 | 256 |
|  | lr | 1e-4 | 1e-4 | 1e-4 |
| Transformer Enc. | $d_{model}$ | 128 | 128 | 128 |
|  | #heads | 2 | 4 | 2 |
|  | #layers | 2 | 2 | 2 |
| Hypernetwork | layers | [128,256] | | |
| Generator | layers | [64,64,64] | [64,64,64,64] | [64,64,64] |
| Random Fourier Features |  | $m = 256, \sigma = 2$ | | |

**For classification with XGBoost**, all hyperparameters used were the default in Chen & Guestrin (2016)'s XGBoost library, with the following exceptions; early stopping was set to 10 rounds, and categorical features were enabled to preserve channel identity as nonordinal.

## A.7 CLASSIFER RESULTS

We present the AUC-ROC scores for different models across varying levels of missingness in Table 11, where higher scores indicate better classification performance.

## A.8 TIMEFLOW RESULTS FOR DIFFERENT MISSINGNESS RATES

To thoroughly demonstrate TV-INRs's capability to handle different missing data scenarios, we conducted extensive experiments by training and testing with various observed ratios ($\tau$), further supporting our claims regarding its efficiency and its ability to serve as a single model for all cases. It is important to note that in the TimeFlow GitHub repository[3], the missing data rate ("draw_ratio")

---

[3] https://github.com/EtienneLnr/TimeFlow/blob/main/experiments/training/
inr_imputation.sh

Table 10: Hyperparameter Grid Search Configuration

| Hyperparameter | Search Range |
|---|---|
| **General Parameters** | |
| Learning rate (lr) | [1e-5, 1e-4, 5e-4] |
| Latent dimension (dim_z) | [16, 32, 64] |
| Dropout rate | [0.0, 0.1, 0.2] |
| **Transformer Encoder** | |
| d_model | [64, 128, 256] |
| Attention layers | [2, 4, 6] |
| Number of heads | [2, 4, 8] |
| Causal attention | [True, False] |
| **Hypernetwork** | |
| Layers | [[32,64], [64,128], [128,256], [256,512]] |
| Activation | ['relu', 'lrelu_01', 'gelu'] |
| **Generator (INR)** | |
| dim_inner | [32,64,128] |
| num_layers | [2, 3, 4] |
| Activation | ['relu', 'lrelu_01', 'gelu'] |
| **Random Fourier Features** | |
| m | [128, 256, 512] |
| $\sigma$ | [1, 2, 4] |

Table 11: AUC-ROC scores for different models across varying levels of missingness. Higher scores indicate better performance. All values are rounded to three decimal places.

| Model | 50% Missingness | 70% Missingness | 95% Missingness |
|---|---|---|---|
| C-TV-INR | 0.969 ± 0.012 | 0.968 ± 0.012 | **0.882 ± 0.028** |
| TV-INR | 0.967 ± 0.013 | 0.963 ± 0.016 | 0.868 ± 0.025 |
| SAITS | 0.906 ± 0.040 | 0.831 ± 0.036 | 0.719 ± 0.039 |
| CSDI | 0.928 ± 0.023 | 0.900 ± 0.035 | 0.847 ± 0.037 |
| Mean Imputation | 0.894 ± 0.039 | 0.818 ± 0.036 | 0.784 ± 0.030 |

can be set as a training argument, with options including $\{0.05, 0.10, 0.20, 0.30, 0.50\}$. Although this may appear to be a hyperparameter choice, it affects the task itself, as the model is optimized for a specific level of missingness.

As shown in Table 12, TimeFlow's performance varies significantly across different training/testing $\tau$ combinations, requiring training different model instances for each scenario. In contrast, TV-INRs has comparable or better performance when compared with Timeflow with a single trained model. These results align with the observation stated in Table 10 of the original TimeFlow paper Naour et al. (2024) that while higher sampling rates simplify the imputation task, they complicate optimization, making it challenging for the model to generalize effectively across different sparsity levels.

## A.9 COMPLEXITY ANALYSIS FOR TV-INR

This section provides the time and memory complexity analysis for the TV-INR model, broken down by its core components: the Transformer-based encoder and the MLP-based decoder (hypernetwork).

**Notation.** To facilitate the analysis, we define the following notation: $L$ is the input sequence length; $C$ is the number of input channels; $E$ is the embedding dimension; $D_p$ is the hidden dimension of the projection layer; $Z$ is the latent dimension; $N$ and $M$ are the number of layers and attention heads in the encoder, respectively; $N'$ and $D_h$ are the number of layers and hidden dimensions of the hypernetwork; and $R$ is the total flattened dimension of the INR parameters being modeled. Typically, the sequence length is the dominant factor, such that $L \gg E \gg Z$.

**Time complexity.** The overall time complexity is determined by the sum of the model's parts. The Transformer-based encoder has a complexity of $O(N \cdot L^2 \cdot E)$, which is quadratic with respect to the sequence length $L$ due to the self-attention mechanism. The subsequent projection layer has a complexity of $O(E \cdot D_p)$. The MLP-based hypernetwork's complexity is $O(Z \cdot D_h + (N' - 1) \cdot D_h^2 + D_h \cdot R)$, which depends on its depth and width. Given that $L$ is the largest dimension, the encoder is the computational bottleneck, making the model's overall time complexity $O(N \cdot L^2 \cdot E)$.

Table 12: **TimeFlow model performance at different training and testing missing ratios ($\tau$).** MSE and MAE metrics are reported for electricity dataset.

| | | | Test $\tau$ | | | | | |
|---|---|---|---|---|---|---|---|---|
| | | | MSE | | | MAE | | |
| Model | $L$ | Train $\tau$ | 0.05 | 0.3 | 0.5 | 0.05 | 0.3 | 0.5 |
| | | 1.00 | 605909.85 | 7.77814 | 0.44302 | 358.39774 | 1.87872 | 0.49501 |
| | | 0.95 | 2611667.2 | 145.28325 | 0.33257 | 587.75934 | 2.32136 | 0.42111 |
| TimeFlow | 200 | 0.50 | 350.9098 | 0.34692 | 0.16299 | 11.31193 | 0.43012 | 0.23984 |
| | | 0.30 | 18.90844 | 0.32993 | 0.20594 | 2.99975 | 0.39625 | 0.30289 |
| | | 0.05 | 0.96294 | 0.74811 | 0.6934 | 0.81073 | 0.71435 | 0.69580 |
| TV-INRs | 200 | $\sim \mathcal{S}$ | 0.3175 | 0.1352 | 0.1132 | 0.3681 | 0.2320 | 0.2123 |
| | | 1.00 | 108812.06 | 0.18195 | 0.13066 | 26.16919 | 0.28272 | 0.25084 |
| | | 0.95 | 22579.357 | 0.15164 | 0.1275 | 15.57548 | 0.27184 | 0.24665 |
| TimeFlow | 2K | 0.50 | 56.5905 | 0.14723 | 0.13238 | 1.88119 | 0.26775 | 0.25275 |
| | | 0.30 | 2.58694 | 0.16536 | 0.15019 | 0.85563 | 0.28756 | 0.27291 |
| | | 0.05 | 0.37793 | 0.22935 | 0.21811 | 0.45838 | 0.34629 | 0.33603 |
| TV-INRs | 2K | $\sim \mathcal{S}$ | 0.2889 | 0.2502 | 0.2491 | 0.3595 | 0.3317 | 0.3311 |

**Memory complexity.** The memory complexity during a forward pass is also dominated by the encoder. The Transformer requires $O(M \cdot L^2)$ memory to store the attention score matrix. The memory requirements for the projection layer and the MLP-based hypernetwork are $O(\max(E, Z))$ and $O(\max(Z, D_h, R))$, respectively, as they are determined by the largest linear layer within each component. Consequently, the overall memory complexity is dictated by the encoder, resulting in $O(M \cdot L^2)$.

## A.10 TRAINING TIMES COMPARISON

In this part, we are reporting the cumulative training times in hours (h) of TV-INRs and Timeflow per task. All training times are rounded to 5-minute intervals and were acquired using an NVIDIA V100 GPU and reported in Tables 13,14,15 and 17,18,19 for imputation and forecasting tasks, respectively. As training times of C-TV-INRs are in the same order with TV-INRs, we omit them to include them in the tables. SAITS demonstrates moderate training times ranging from 1h45m to 13h35m across various datasets, offering a reasonable compromise between efficiency and performance. A drawback of CSDI Tashiro et al. (2021) is its extended training duration, primarily due to the iterative optimization process inherent in diffusion model training. DeepTime Woo et al. (2023) is very fast to train due to number of epochs selected in the original work; however it also has the worst performance among the baselines as shown in Table 2. Our primary baseline, TimeFlow, demands significantly greater computational resources, with cumulative training durations consistently exceeding those of TV-INR across most experimental scenarios. Efficiency analyses reveal TimeFlow requires up to 3.70× longer training periods, particularly pronounced in forecasting applications as shown in Table 20.

Table 13: Training times for imputation task, TV-INRs.

| Model Name | Dataset | L | Max Epochs | Training Time |
|---|---|---|---|---|
| TV-INR | Electricity | 200 | 2000 | 8h45m |
| TV-INR | Electricity | 2000 | 4000 | 12h55m |
| TV-INR | Traffic | 200 | 2000 | 10h35m |
| TV-INR | Traffic | 2000 | 4000 | 15h50m |
| TV-INR | Solar-10 | 200 | 2000 | 10h25m |
| TV-INR | Solar-10 | 10000 | 4000 | 19h15m |
| TV-INR | HAR | 128 | 3000 | 6h45m |
| TV-INR | P12 | 128 | 1000 | 4h05m |

Table 14: Training times for imputation task, TimeFlow.

| Model Name | Dataset | L | $\tau$ | Max Epochs | Training Time |
|---|---|---|---|---|---|
| TimeFlow | Electricity | 200 | 0.05 | 40000 | 6h35m |
| TimeFlow | Electricity | 200 | 0.30 | 40000 | 6h40m |
| TimeFlow | Electricity | 200 | 0.50 | 40000 | 6h35m |
| TimeFlow | Electricity | 2000 | 0.05 | 40000 | 5h35m |
| TimeFlow | Electricity | 2000 | 0.30 | 40000 | 5h30m |
| TimeFlow | Electricity | 2000 | 0.50 | 40000 | 5h40m |
| TimeFlow | Traffic | 200 | 0.05 | 40000 | 9h45m |
| TimeFlow | Traffic | 200 | 0.30 | 40000 | 9h50m |
| TimeFlow | Traffic | 200 | 0.50 | 40000 | 10h10m |
| TimeFlow | Traffic | 2000 | 0.05 | 40000 | 8h30m |
| TimeFlow | Traffic | 2000 | 0.30 | 40000 | 8h30m |
| TimeFlow | Traffic | 2000 | 0.50 | 40000 | 8h45m |
| TimeFlow | Solar-10 | 200 | 0.05 | 40000 | 6h45m |
| TimeFlow | Solar-10 | 200 | 0.30 | 40000 | 6h30m |
| TimeFlow | Solar-10 | 200 | 0.50 | 40000 | 6h35m |
| TimeFlow | Solar-10 | 10000 | 0.05 | 40000 | 12h5m |
| TimeFlow | Solar-10 | 10000 | 0.30 | 40000 | 11h50m |
| TimeFlow | Solar-10 | 10000 | 0.50 | 40000 | 12h15m |

Table 15: Training times for imputation task, SAITS.

| Model Name | Dataset | L | Max Epochs | Training Time |
|---|---|---|---|---|
| SAITS | Electricity | 200 | 10000 | 3h45m |
| SAITS | Electricity | 2000 | 10000 | 3h35m |
| SAITS | Traffic | 200 | 10000 | 3h25m |
| SAITS | Traffic | 2000 | 10000 | 7h45m |
| SAITS | Solar-10 | 200 | 10000 | 1h45m |
| SAITS | Solar-10 | 10000 | 10000 | 6h05m |
| SAITS | HAR | 128 | 10000 | 13h35m |
| SAITS | P12 | 48 | 10000 | 10h40m |

Table 16: Training times for imputation task, CSDI.

| Model Name | Dataset | L | Max Epochs | Training Time |
|---|---|---|---|---|
| CSDI | Electricity | 200 | 200 | 2h55m |
| CSDI | Electricity | 2000 | 200 | 6h |
| CSDI | Traffic | 200 | 200 | 3h20m |
| CSDI | Traffic | 2000 | 200 | 7h20m |
| CSDI | Solar-10 | 200 | 200 | 1h30m |
| CSDI | Solar-10 | 10000 | 200 | 12h |
| CSDI | HAR | 128 | 200 | 8h5m |
| CSDI | P12 | 48 | 200 | 16h10m |

Table 17: Training times for forecasting task, TV-INRs.

| Model Name | Dataset | H | Max Epochs | Training Time |
|---|---|---|---|---|
| TV-INR | Electricity | 512 | 2000 | 5h25m |
| TV-INR | Traffic | 512 | 4000 | 11h05m |
| TV-INR | Solar-H | 512 | 2000 | 5h15m |

Table 18: Training times for forecasting task, TimeFlow.

| Model Name | Dataset | H | F | Max Epochs | Training Time |
|---|---|---|---|---|---|
| TimeFlow | Electricity | 512 | 96 | 40000 | 4h25m |
| TimeFlow | Electricity | 512 | 192 | 40000 | 4h30m |
| TimeFlow | Electricity | 512 | 336 | 40000 | 4h40m |
| TimeFlow | Electricity | 512 | 720 | 40000 | 4h30m |
| TimeFlow | Traffic | 512 | 96 | 40000 | 10h10m |
| TimeFlow | Traffic | 512 | 192 | 40000 | 10h15m |
| TimeFlow | Traffic | 512 | 336 | 40000 | 10h20m |
| TimeFlow | Traffic | 512 | 720 | 40000 | 10h15m |
| TimeFlow | Solar-H | 512 | 96 | 40000 | 3h25m |
| TimeFlow | Solar-H | 512 | 192 | 40000 | 2h55m |
| TimeFlow | Solar-H | 512 | 336 | 40000 | 3h05m |
| TimeFlow | Solar-H | 512 | 720 | 40000 | 3h15m |

Table 19: Training times for forecasting task, DeepTime.

| Model Name | Dataset | H | F | Max Epochs | Training Time |
|---|---|---|---|---|---|
| DeepTime | Electricity | 512 | 96 | 50 | 5m |
| DeepTime | Electricity | 512 | 192 | 50 | 5m |
| DeepTime | Electricity | 512 | 336 | 50 | 5m |
| DeepTime | Electricity | 512 | 720 | 50 | 10m |
| DeepTime | Traffic | 512 | 96 | 50 | 10m |
| DeepTime | Traffic | 512 | 192 | 50 | 10m |
| DeepTime | Traffic | 512 | 336 | 50 | 15m |
| DeepTime | Traffic | 512 | 720 | 50 | 15m |
| DeepTime | Solar-H | 512 | 96 | 50 | 5m |
| DeepTime | Solar-H | 512 | 192 | 50 | 5m |
| DeepTime | Solar-H | 512 | 336 | 50 | 5m |
| DeepTime | Solar-H | 512 | 720 | 50 | 5m |

Table 20: **Training Time Efficiency Ratio: TV-INR vs TimeFlow** in hours (h).

| Forecasting Task | | TV-INR | TimeFlow | Ratio (TimeFlow/TV-INR) | |
|---|---|---|---|---|---|
| Dataset | $H$ | Training Time (h) | Cumulative Time (h) | Absolute | Multiplier |
| Electricity | 512 | 5.42 | 18.08 | 12.66 | 3.34× |
| Traffic | 512 | 11.08 | 41.00 | 29.92 | 3.70× |
| Solar | 512 | 5.25 | 12.67 | 7.42 | 2.41× |
| Imputation Task | | TV-INR | TimeFlow | Ratio (TimeFlow/TV-INR) | |
| Dataset | $L$ | Training Time (h) | Cumulative Time (h) | Absolute | Multiplier |
| Electricity | 200 | 8.75 | 19.83 | 11.08 | 2.27× |
| Electricity | 2000 | 12.92 | 16.75 | 3.83 | 1.30× |
| Traffic | 200 | 10.58 | 29.75 | 19.17 | 2.81× |
| Traffic | 2000 | 15.83 | 25.75 | 9.92 | 1.63× |
| Solar | 200 | 10.42 | 19.83 | 9.41 | 1.90× |
| Solar | 10000 | 19.25 | 36.17 | 16.92 | 1.88× |

## A.11 INFERENCE TIMES COMPARISON

We evaluated the computational efficiency of TV-INRs against TimeFlow by measuring inference times on an NVIDIA V100 GPU. Under identical conditions with a batch size of 1, we recorded forward pass execution times in seconds for both models. TimeFlow was configured to use 3 gradient steps during meta-learning, as specified in the original paper Naour et al. (2024). A key advantage of TV-INRs is that its inference time remains constant, unlike TimeFlow, which exhibits linear scaling with the number of gradient steps performed during meta-learning. This makes TV-INRs particularly attractive for applications requiring consistent and predictable inference latency.

Table 21: Comparison of inference time of TV-INRs and SAITS in seconds for imputation task.

| Model | $L$ | $\tau_{\text{Train}}$ | $\tau_{\text{Test}}$ | Electricity Time (s) | Traffic Time (s) | $L$ | Solar-10 Time (s) |
|-------|-----|------------------------|----------------------|----------------------|------------------|-----|-------------------|
| TimeFlow | 2K | 0.50 | 0.50 | $0.017 \pm 0.001$ | $0.016 \pm 0.001$ | | $0.038 \pm 0.001$ |
| | | 0.30 | 0.30 | $0.016 \pm 0.001$ | $0.016 \pm 0.001$ | 10K | $0.037 \pm 0.001$ |
| | | 0.05 | 0.05 | $0.016 \pm 0.001$ | $0.016 \pm 0.001$ | | $0.037 \pm 0.001$ |
| TimeFlow | 200 | 0.50 | 0.50 | $0.013 \pm 0.001$ | $0.015 \pm 0.001$ | | $0.015 \pm 0.001$ |
| | | 0.30 | 0.30 | $0.012 \pm 0.001$ | $0.015 \pm 0.001$ | 200 | $0.015 \pm 0.001$ |
| | | 0.05 | 0.05 | $0.012 \pm 0.001$ | $0.015 \pm 0.001$ | | $0.015 \pm 0.001$ |
| TV-INRs | 2K | $\sim \mathcal{S}$ | 0.50 | $0.016 \pm 0.001$ | $0.017 \pm 0.001$ | | $0.060 \pm 0.001$ |
| | | | 0.30 | $0.017 \pm 0.001$ | $0.017 \pm 0.001$ | 10K | $0.059 \pm 0.001$ |
| | | | 0.05 | $0.017 \pm 0.001$ | $0.017 \pm 0.001$ | | $0.059 \pm 0.001$ |
| TV-INRs | 200 | $\sim \mathcal{S}$ | 0.50 | $0.014 \pm 0.001$ | $0.013 \pm 0.001$ | | $0.014 \pm 0.001$ |
| | | | 0.30 | $0.014 \pm 0.002$ | $0.013 \pm 0.001$ | 200 | $0.014 \pm 0.001$ |
| | | | 0.05 | $0.014 \pm 0.001$ | $0.013 \pm 0.001$ | | $0.014 \pm 0.001$ |

Table 22: Comparison of inference time of TV-INRs and Timeflow in seconds for forecasting task.

| Model | H | $F_{\text{train}}$ | $F_{\text{test}}$ | Electricity Time (s) | Traffic Time (s) | Solar-H Time (s) |
|-------|---|--------------------|--------------------|----------------------|------------------|------------------|
| TimeFlow | 512 | 96 | 96 | $0.016 \pm 0.001$ | $0.017 \pm 0.001$ | $0.016 \pm 0.001$ |
| | | 192 | 192 | $0.016 \pm 0.001$ | $0.019 \pm 0.001$ | $0.015 \pm 0.001$ |
| | | 336 | 336 | $0.016 \pm 0.001$ | $0.020 \pm 0.001$ | $0.015 \pm 0.001$ |
| | | 720 | 720 | $0.016 \pm 0.001$ | $0.020 \pm 0.001$ | $0.015 \pm 0.001$ |
| TV-INRs | 512 | $\sim \mathcal{F}$ | 720 | $0.016 \pm 0.001$ | $0.018 \pm 0.001$ | $0.017 \pm 0.002$ |

## A.12 ABLATION STUDY ON THE NUMBER OF FOURIER FREQUENCIES

To empirically quantify the contribution of Fourier Features to the performance of TV-INR, we conduct an ablation study analyzing the model's performance with different numbers of Fourier frequencies ($N_{\text{FF}}$). The experiment is conducted on Electricity dataset for imputation task, and the results are reported, with performance statistics—mean and standard deviation—computed over multiple non-overlapping test windows. The table below presents the Mean Squared Error (MSE) on the imputed values for configurations with $N_{\text{FF}} \in \{256, 128, 32, 0\}$. The results clearly demonstrate that incorporating Fourier Features provides a significant performance benefit, which aligns with findings in the broader literature Tancik et al. (2020); Dupont et al. (2021). Across all sequence lengths and observation rates, performance degrades substantially as the number of frequencies is reduced, with the best results consistently achieved for $N_{\text{FF}} = 256$.

Table 23: Ablation study on the effect of Fourier Features. We report MSE on the Electricity dataset for different numbers of Fourier Feature frequencies ($N_{\text{FF}}$). The best performing configuration for each row is in bold.

| Model | L | $\tau$ | Number of Fourier Feature Frequencies ($N_{\text{FF}}$) 256 | 128 | 32 | 0 (None) |
|-------|---|--------|------|-----|-----|----------|
| TV-INRs | 200 | 0.50 | $\mathbf{0.1213 \pm 0.0131}$ | $0.1391 \pm 0.0140$ | $0.1523 \pm 0.0186$ | $0.8099 \pm 0.0522$ |
| | | 0.30 | $\mathbf{0.1359 \pm 0.0265}$ | $0.1756 \pm 0.0211$ | $0.2711 \pm 0.0386$ | $0.8587 \pm 0.0502$ |
| | | 0.05 | $\mathbf{0.3312 \pm 0.0968}$ | $0.4655 \pm 0.1198$ | $0.8643 \pm 0.1206$ | $1.2215 \pm 0.1335$ |
| TV-INRs | 2000 | 0.50 | $\mathbf{0.2555 \pm 0.0280}$ | $0.3563 \pm 0.0236$ | $1.0414 \pm 0.0233$ | $1.0542 \pm 0.0239$ |
| | | 0.30 | $\mathbf{0.2423 \pm 0.0276}$ | $0.3444 \pm 0.0095$ | $1.0341 \pm 0.0503$ | $1.0531 \pm 0.0221$ |
| | | 0.05 | $\mathbf{0.3142 \pm 0.0742}$ | $0.4984 \pm 0.0390$ | $1.0687 \pm 0.0400$ | $1.1004 \pm 0.0278$ |

## A.13 COMPARISON WITH STANDARD VAE BASELINE

To empirically validate the contribution of our Implicit Neural Representation (INR) based decoder, we conduct an ablation study comparing TV-INR against a baseline with a standard decoder, which

we term TV-VAE. This baseline is designed to isolate the impact of the INR by replacing the hypernetwork decoder with a conventional MLP. Specifically, the TV-VAE decoder processes a direct concatenation of the learned latent representation $z$ and the time encoding $t$. To ensure a fair comparison, the MLP architecture for the TV-VAE decoder is constructed from the same building blocks as the hypernetwork in TV-INR.

We performed a thorough hyperparameter search for the TV-VAE model, evaluating various MLP depths and multiple configurations of Fourier Features for the time encoding. All other experimental settings, including the AdamW optimizer, followed the protocol used for the main TV-INR experiments as detailed in App. A.6. The results, presented in App. [Reference to the new tables], show that TV-INR consistently and significantly outperforms all tested variants of TV-VAE on the electricity dataset for sequence lengths $L = 200, 2000$ and across all observation rates ($\tau$). This consistent superiority demonstrates that the INR-based architecture is more effective at modeling the continuous temporal structure of time series signals than a standard decoder that treats time as a concatenated input feature, thereby justifying our architectural choice.

Table 24: Ablation study on the **Electricity dataset (L=200)**. We compare TV-INR with TV-VAE variants using different MLP decoder depths ($D$) and numbers of Fourier Feature frequencies ($N_{\text{FF}}$). Best results are in bold.

| Model | $D$ | $N_{\text{FF}}$ | $\tau = 0.05$ | | $\tau = 0.3$ | | $\tau = 0.5$ | |
|---|---|---|---|---|---|---|---|---|
| | | | MSE | MAE | MSE | MAE | MSE | MAE |
| TV-VAE | 5 | 256 | $0.98 \pm 0.22$ | $0.78 \pm 0.10$ | $0.44 \pm 0.10$ | $0.48 \pm 0.06$ | $0.34 \pm 0.07$ | $0.41 \pm 0.05$ |
| TV-VAE | 5 | 128 | $1.00 \pm 0.21$ | $0.80 \pm 0.01$ | $0.48 \pm 0.12$ | $0.51 \pm 0.08$ | $0.35 \pm 0.08$ | $0.42 \pm 0.05$ |
| TV-VAE | 5 | 32 | $1.11 \pm 0.39$ | $0.83 \pm 0.16$ | $0.52 \pm 0.16$ | $0.52 \pm 0.09$ | $0.36 \pm 0.10$ | $0.42 \pm 0.06$ |
| TV-VAE | 5 | 0 | $1.24 \pm 0.14$ | $0.83 \pm 0.06$ | $0.52 \pm 0.05$ | $0.50 \pm 0.02$ | $0.43 \pm 0.05$ | $0.45 \pm 0.02$ |
| TV-VAE | 4 | 256 | $0.90 \pm 0.14$ | $0.74 \pm 0.07$ | $0.32 \pm 0.05$ | $0.39 \pm 0.04$ | $0.23 \pm 0.04$ | $0.33 \pm 0.03$ |
| TV-VAE | 4 | 128 | $1.07 \pm 0.14$ | $0.84 \pm 0.06$ | $0.57 \pm 0.08$ | $0.59 \pm 0.05$ | $0.43 \pm 0.07$ | $0.51 \pm 0.04$ |
| TV-VAE | 4 | 32 | $0.65 \pm 0.12$ | $0.61 \pm 0.07$ | $0.25 \pm 0.04$ | $0.34 \pm 0.03$ | $0.20 \pm 0.04$ | $0.30 \pm 0.02$ |
| TV-VAE | 4 | 0 | $1.41 \pm 0.11$ | $0.91 \pm 0.04$ | $0.59 \pm 0.10$ | $0.54 \pm 0.05$ | $0.45 \pm 0.07$ | $0.47 \pm 0.03$ |
| TV-VAE | 3 | 256 | $0.62 \pm 0.16$ | $0.59 \pm 0.08$ | $0.21 \pm 0.04$ | $0.31 \pm 0.03$ | $0.18 \pm 0.03$ | $0.28 \pm 0.02$ |
| TV-VAE | 3 | 128 | $0.50 \pm 0.12$ | $0.43 \pm 0.07$ | $0.19 \pm 0.04$ | $0.28 \pm 0.03$ | $0.17 \pm 0.03$ | $0.27 \pm 0.02$ |
| TV-VAE | 3 | 32 | $0.66 \pm 0.13$ | $0.62 \pm 0.08$ | $0.25 \pm 0.05$ | $0.34 \pm 0.03$ | $0.20 \pm 0.03$ | $0.30 \pm 0.02$ |
| TV-VAE | 3 | 0 | $1.58 \pm 0.27$ | $0.97 \pm 0.08$ | $0.63 \pm 0.09$ | $0.59 \pm 0.04$ | $0.51 \pm 0.06$ | $0.53 \pm 0.03$ |
| TV-VAE | 2 | 256 | $0.88 \pm 0.13$ | $0.78 \pm 0.07$ | $0.45 \pm 0.06$ | $0.53 \pm 0.05$ | $0.34 \pm 0.06$ | $0.44 \pm 0.04$ |
| TV-VAE | 2 | 128 | $0.87 \pm 0.12$ | $0.78 \pm 0.06$ | $0.41 \pm 0.05$ | $0.51 \pm 0.04$ | $0.30 \pm 0.05$ | $0.42 \pm 0.04$ |
| TV-VAE | 2 | 32 | $0.79 \pm 0.20$ | $0.70 \pm 0.10$ | $0.30 \pm 0.05$ | $0.40 \pm 0.04$ | $0.23 \pm 0.04$ | $0.34 \pm 0.03$ |
| TV-VAE | 2 | 0 | $1.59 \pm 0.51$ | $0.97 \pm 0.11$ | $0.84 \pm 0.08$ | $0.71 \pm 0.03$ | $0.76 \pm 0.08$ | $0.67 \pm 0.03$ |
| TV-VAE | 1 | 256 | $0.39 \pm 0.10$ | $0.43 \pm 0.07$ | $0.21 \pm 0.05$ | $0.30 \pm 0.03$ | $0.20 \pm 0.04$ | $0.29 \pm 0.03$ |
| TV-VAE | 1 | 128 | $0.41 \pm 0.06$ | $0.43 \pm 0.08$ | $0.21 \pm 0.05$ | $0.30 \pm 0.03$ | $0.20 \pm 0.04$ | $0.30 \pm 0.03$ |
| TV-VAE | 1 | 32 | $0.39 \pm 0.06$ | $0.44 \pm 0.05$ | $0.23 \pm 0.05$ | $0.32 \pm 0.03$ | $0.22 \pm 0.04$ | $0.31 \pm 0.03$ |
| TV-VAE | 1 | 0 | $1.37 \pm 0.12$ | $0.93 \pm 0.04$ | $1.13 \pm 0.05$ | $0.84 \pm 0.02$ | $1.09 \pm 0.07$ | $0.82 \pm 0.02$ |
| TV-INRs | 3 | 256 | $\mathbf{0.32 \pm 0.06}$ | $\mathbf{0.37 \pm 0.04}$ | $\mathbf{0.14 \pm 0.03}$ | $\mathbf{0.23 \pm 0.02}$ | $\mathbf{0.11 \pm 0.02}$ | $\mathbf{0.21 \pm 0.02}$ |

Table 25: Ablation study on the **Electricity dataset (L=2000)**. We compare TV-INR with TV-VAE variants using different MLP decoder depths ($D$) and numbers of Fourier Feature frequencies ($N_{FF}$). Best results are in bold.

| Model | $D$ | $N_{FF}$ | $\tau = 0.05$ | | $\tau = 0.3$ | | $\tau = 0.5$ | |
|---|---|---|---|---|---|---|---|---|
| | | | MSE | MAE | MSE | MAE | MSE | MAE |
| TV-VAE | 6 | 256 | $0.92 \pm 0.11$ | $0.78 \pm 0.05$ | $0.51 \pm 0.04$ | $0.51 \pm 0.03$ | $0.43 \pm 0.03$ | $0.46 \pm 0.02$ |
| TV-VAE | 6 | 128 | $0.43 \pm 0.06$ | $0.46 \pm 0.03$ | $0.37 \pm 0.04$ | $0.42 \pm 0.03$ | $0.36 \pm 0.04$ | $0.42 \pm 0.03$ |
| TV-VAE | 6 | 32 | $0.94 \pm 0.02$ | $0.74 \pm 0.01$ | $0.89 \pm 0.03$ | $0.71 \pm 0.01$ | $0.89 \pm 0.02$ | $0.71 \pm 0.01$ |
| TV-VAE | 6 | 0 | $1.17 \pm 0.03$ | $0.84 \pm 0.01$ | $1.06 \pm 0.02$ | $0.80 \pm 0.01$ | $1.06 \pm 0.02$ | $0.80 \pm 0.01$ |
| TV-VAE | 5 | 256 | $1.06 \pm 0.23$ | $0.83 \pm 0.11$ | $0.61 \pm 0.07$ | $0.59 \pm 0.05$ | $0.46 \pm 0.03$ | $0.48 \pm 0.02$ |
| TV-VAE | 5 | 128 | $0.44 \pm 0.05$ | $0.46 \pm 0.04$ | $0.38 \pm 0.04$ | $0.43 \pm 0.03$ | $0.37 \pm 0.04$ | $0.42 \pm 0.02$ |
| TV-VAE | 5 | 32 | $0.92 \pm 0.03$ | $0.72 \pm 0.01$ | $0.86 \pm 0.03$ | $0.70 \pm 0.01$ | $0.86 \pm 0.03$ | $0.70 \pm 0.01$ |
| TV-VAE | 5 | 0 | $1.16 \pm 0.03$ | $0.84 \pm 0.01$ | $1.05 \pm 0.02$ | $0.80 \pm 0.01$ | $1.05 \pm 0.02$ | $0.80 \pm 0.01$ |
| TV-VAE | 4 | 256 | $0.33 \pm 0.02$ | $0.39 \pm 0.02$ | $0.28 \pm 0.02$ | $0.36 \pm 0.01$ | $0.26 \pm 0.02$ | $0.35 \pm 0.01$ |
| TV-VAE | 4 | 128 | $0.35 \pm 0.03$ | $0.41 \pm 0.02$ | $0.32 \pm 0.02$ | $0.39 \pm 0.01$ | $0.32 \pm 0.02$ | $0.39 \pm 0.01$ |
| TV-VAE | 4 | 32 | $0.75 \pm 0.02$ | $0.67 \pm 0.02$ | $0.72 \pm 0.02$ | $0.65 \pm 0.02$ | $0.72 \pm 0.03$ | $0.65 \pm 0.02$ |
| TV-VAE | 4 | 0 | $1.10 \pm 0.01$ | $0.83 \pm 0.01$ | $1.04 \pm 0.02$ | $0.80 \pm 0.01$ | $1.05 \pm 0.02$ | $0.80 \pm 0.01$ |
| TV-VAE | 3 | 256 | $0.37 \pm 0.02$ | $0.43 \pm 0.02$ | $0.33 \pm 0.02$ | $0.40 \pm 0.02$ | $0.32 \pm 0.03$ | $0.40 \pm 0.02$ |
| TV-VAE | 3 | 128 | $0.43 \pm 0.05$ | $0.48 \pm 0.04$ | $0.40 \pm 0.04$ | $0.46 \pm 0.03$ | $0.40 \pm 0.04$ | $0.46 \pm 0.03$ |
| TV-VAE | 3 | 32 | $0.98 \pm 0.01$ | $0.80 \pm 0.01$ | $0.91 \pm 0.01$ | $0.77 \pm 0.01$ | $0.91 \pm 0.01$ | $0.76 \pm 0.01$ |
| TV-VAE | 3 | 0 | $1.09 \pm 0.01$ | $0.82 \pm 0.01$ | $1.04 \pm 0.03$ | $0.80 \pm 0.01$ | $1.05 \pm 0.02$ | $0.80 \pm 0.01$ |
| TV-VAE | 2 | 256 | $0.34 \pm 0.03$ | $0.41 \pm 0.02$ | $0.31 \pm 0.02$ | $0.38 \pm 0.01$ | $0.30 \pm 0.02$ | $0.38 \pm 0.01$ |
| TV-VAE | 2 | 128 | $0.56 \pm 0.08$ | $0.58 \pm 0.05$ | $0.53 \pm 0.07$ | $0.56 \pm 0.05$ | $0.53 \pm 0.08$ | $0.56 \pm 0.05$ |
| TV-VAE | 2 | 32 | $1.05 \pm 0.01$ | $0.82 \pm 0.01$ | $1.01 \pm 0.03$ | $0.79 \pm 0.01$ | $1.01 \pm 0.02$ | $0.79 \pm 0.01$ |
| TV-VAE | 2 | 0 | $1.08 \pm 0.01$ | $0.81 \pm 0.01$ | $1.06 \pm 0.03$ | $0.80 \pm 0.01$ | $1.06 \pm 0.02$ | $0.80 \pm 0.01$ |
| TV-INRs | 4 | 256 | $\mathbf{0.29 \pm 0.02}$ | $\mathbf{0.36 \pm 0.02}$ | $\mathbf{0.25 \pm 0.02}$ | $\mathbf{0.33 \pm 0.01}$ | $\mathbf{0.25 \pm 0.02}$ | $\mathbf{0.33 \pm 0.01}$ |

## B  APPENDIX B

### B.1  VISUALS FROM EXPERIMENTS

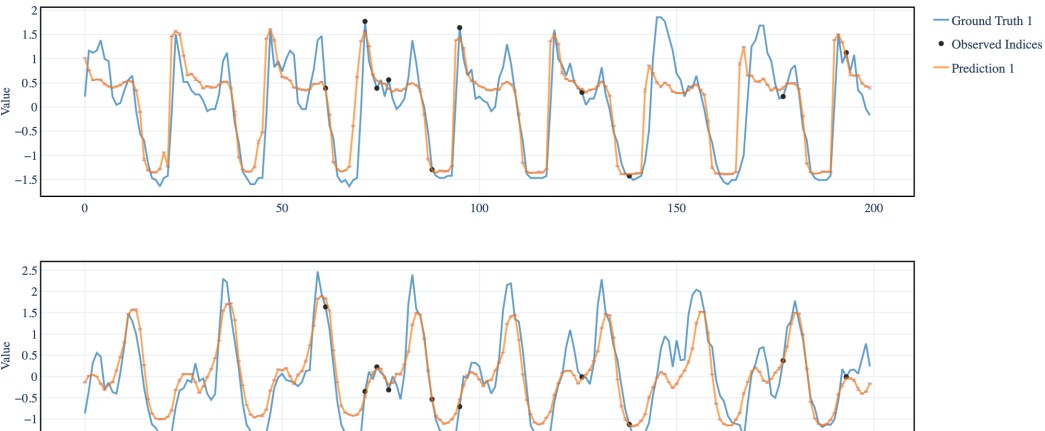

(a) Imputation task for Electricity dataset $L = 200$, $\tau = 0.05$.

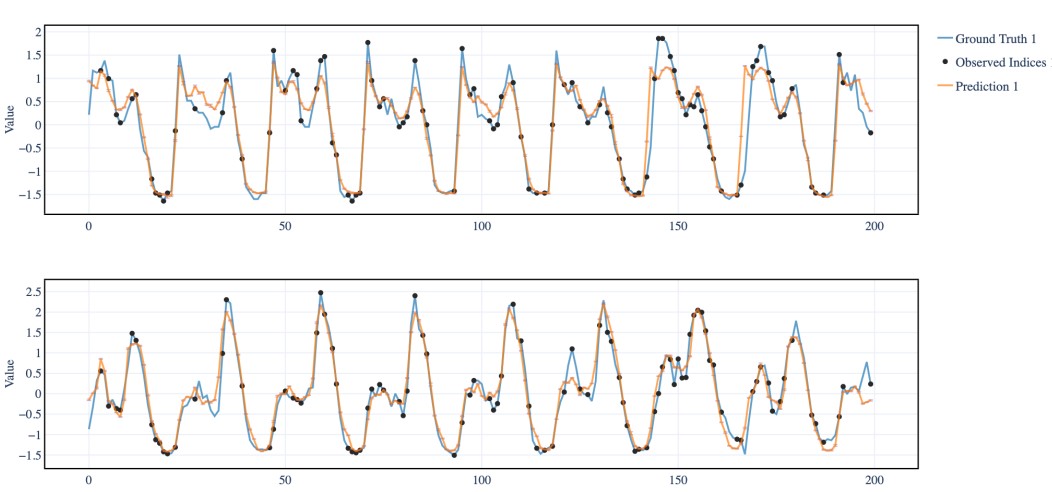

(b) Imputation task for Electricity dataset $L = 200$, $\tau = 0.5$.

Figure 4: TV-INRs imputation predictions for Electricity dataset ($L = 200$).

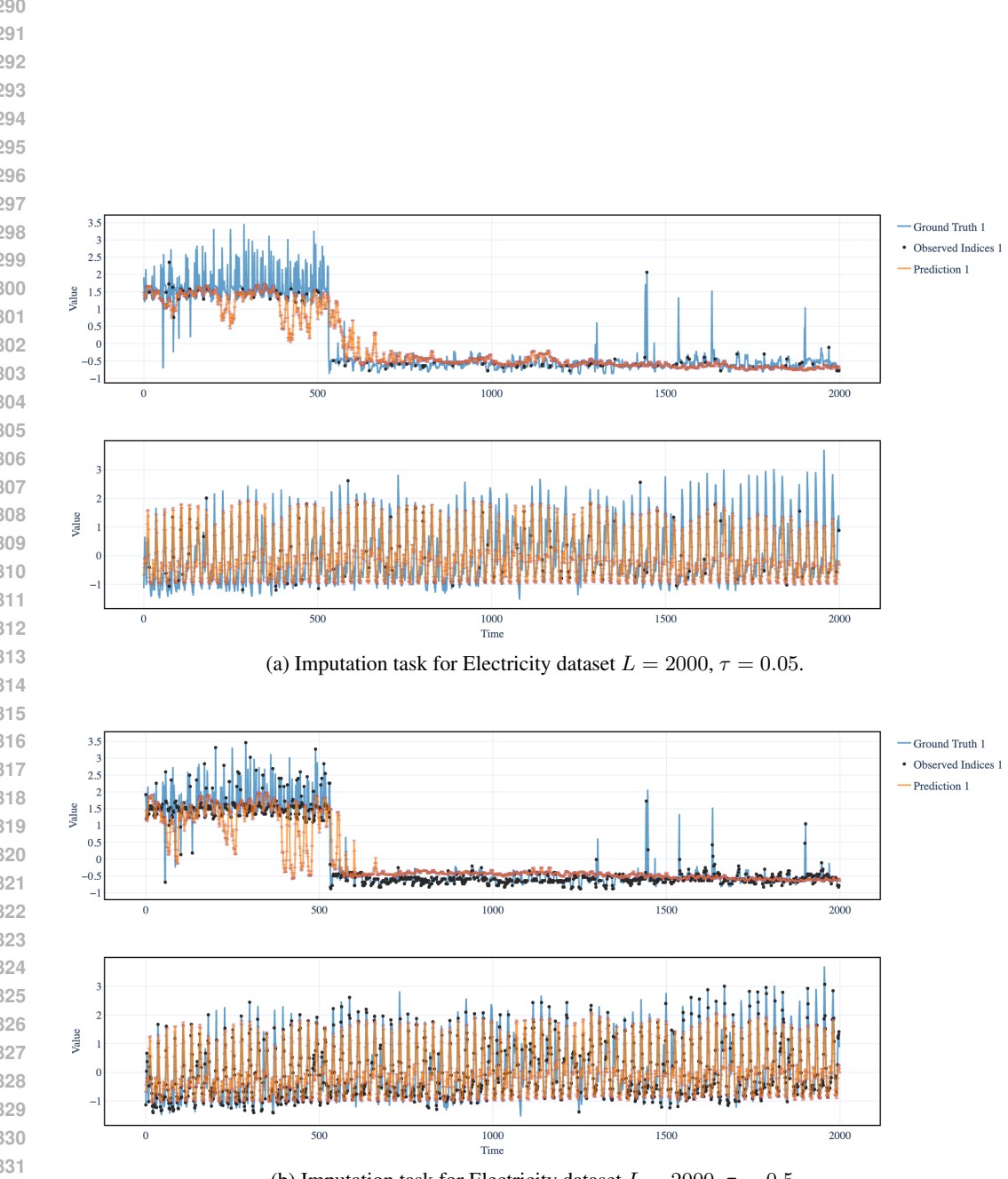

(a) Imputation task for Electricity dataset $L = 2000$, $\tau = 0.05$.

(b) Imputation task for Electricity dataset $L = 2000$, $\tau = 0.5$.

Figure 5: TV-INRs imputation predictions for Electricity dataset ($L = 2000$).

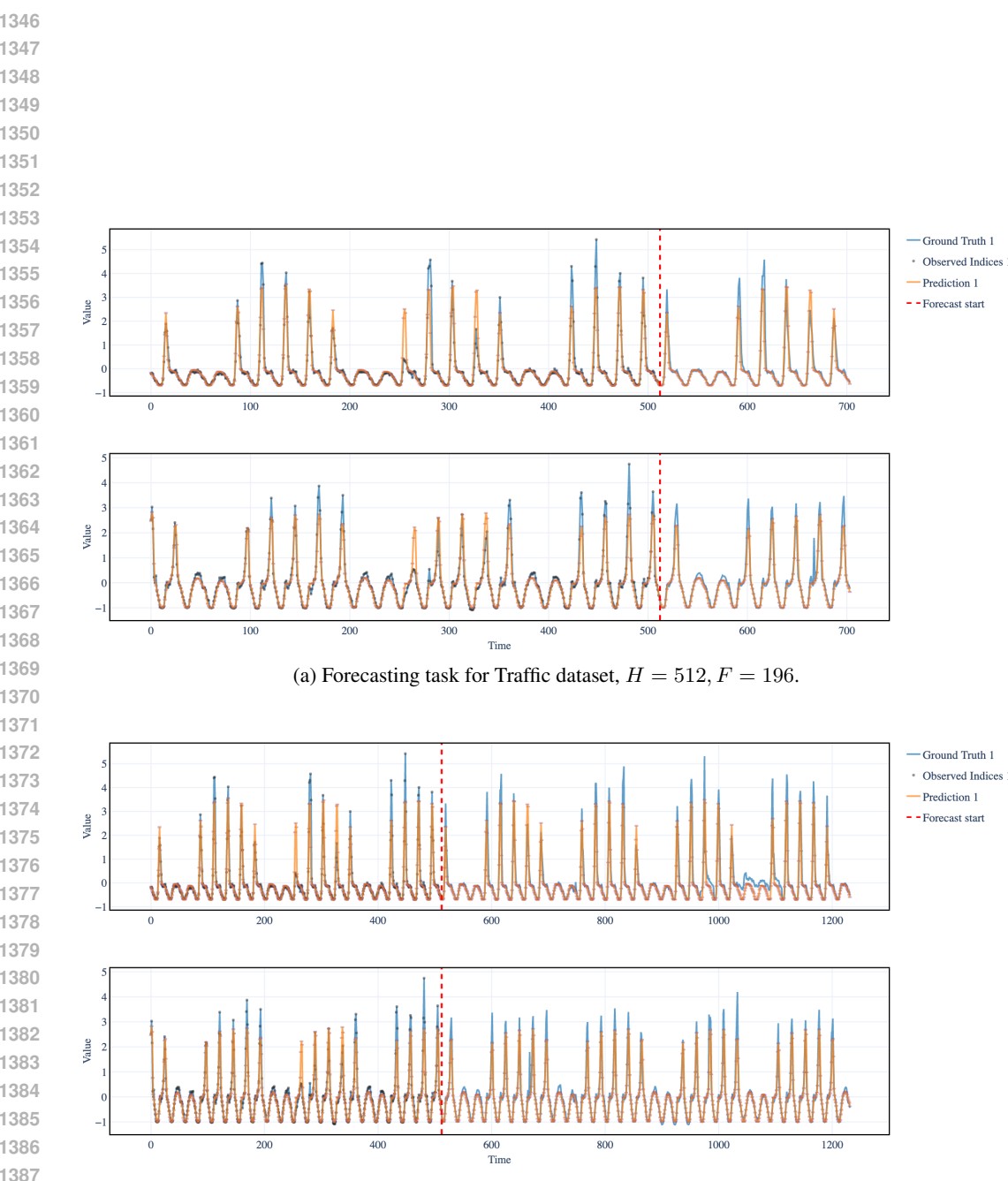

(a) Forecasting task for Traffic dataset, $H = 512, F = 196$.

(b) Forecasting task for Traffic dataset, $H = 512, F = 720$.

Figure 6: TV-INRs forecasting predictions for Traffic dataset.

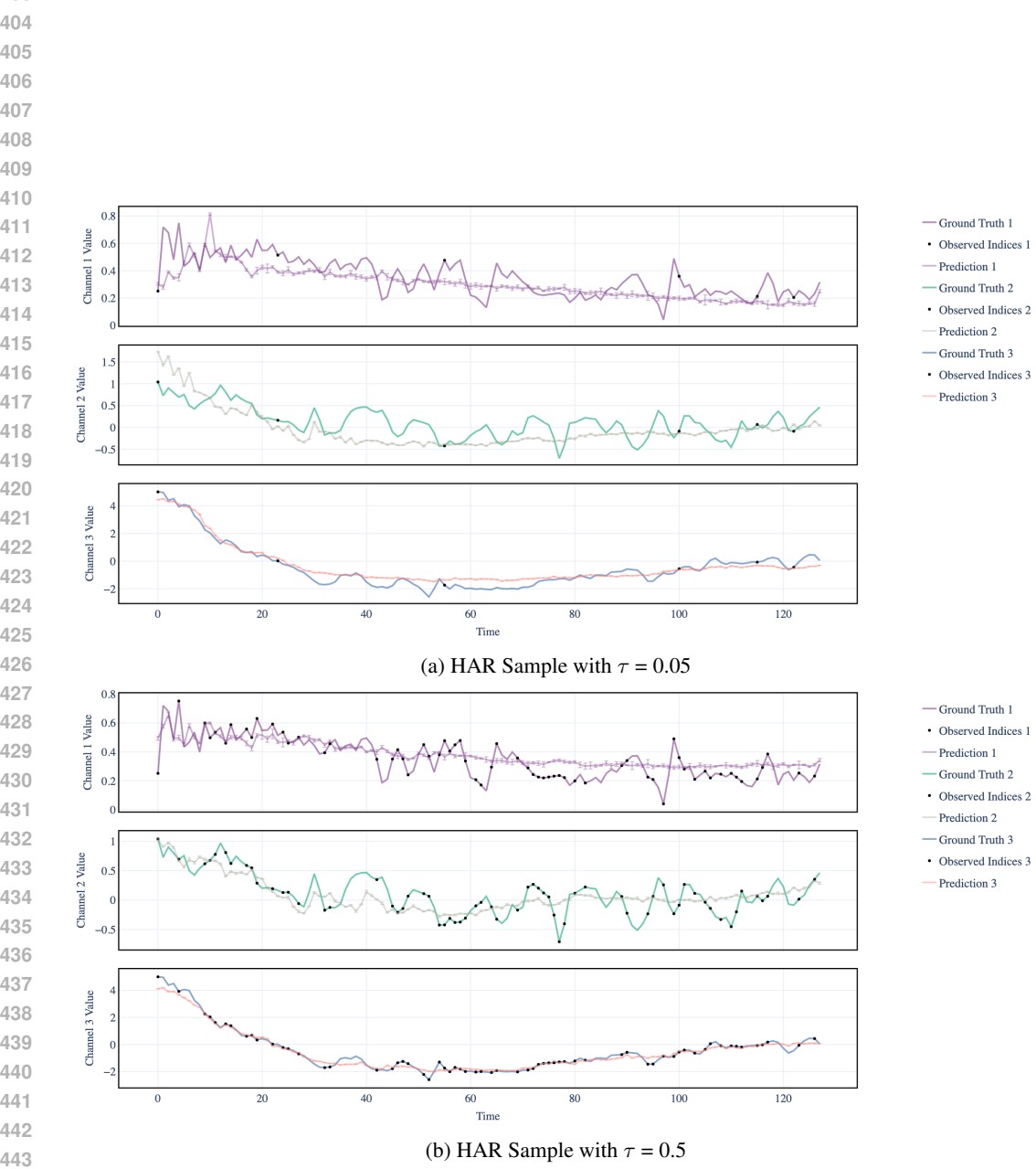

(a) HAR Sample with $\tau = 0.05$

(b) HAR Sample with $\tau = 0.5$

Figure 7: TV-INRs imputations for HAR dataset.

