# OpenReview forum: "Temporal Variational Implicit Neural Representations"
_ICLR.cc/2026/Conference — ICLR 2026 Conference Withdrawn Submission_

### Official Review · Reviewer_34rW · 2025-10-18

**Soundness:** 3
**Presentation:** 3
**Contribution:** 2
**Rating:** 4
**Confidence:** 4

**Summary:**

This paper introduces Temporal Variational Implicit Neural Representations (TV-INRs), a probabilistic framework for modeling irregular and multivariate time series. The method combines Implicit Neural Representations (INRs) with a VAE-based hypernetwork to learn distributions over latent time series representations.

Specifically, (i) observed timestamp–value pairs are encoded through a Transformer encoder whose outputs parameterize a Gaussian distribution, and (ii) a latent variable is sampled from this distribution and decoded—optionally conditioned on covariate information—to generate the INR weights. This encoding–decoding process enables individualized imputation and forecasting without requiring meta-learning or per-sample optimization.

Experiments are conducted on three univariate datasets for imputation and forecasting and two multivariate datasets for imputation. The proposed model demonstrates strong performance in low-data and sparse regimes

**Strengths:**

1. The choice to use Implicit Neural Representations (INRs) for time series modeling — including imputation, forecasting, and handling irregular sampling — is elegant. The paper is generally well-written, and the appendices are detailed and comprehensive.
2. Compared to prior INR-based approaches for time series, the proposed model is naturally suited to handle covariates and multivariate data, which is a clear improvement.
3. The ability to handle varying levels of missingness (in imputation tasks, for example) efficiently — what the authors call “generalization” — without retraining the model is a good point for real-world applications.
4. In specific settings, such as short time series or extremely sparse observations, the experiments show that TV-INR can outperform competitive models.

**Weaknesses:**

1. The paper’s first main claim is the introduction of a probabilistic framework for INR-based time series modeling, but this aspect is neither clearly motivated nor actually leveraged in the experiments. It could have been justified through, for example, quantile prediction and uncertainty estimation metrics through the WQL metric.
2. The second claim concerns inference “without requiring per-sample optimization,” in contrast to meta-learning methods like TimeFlow or DeepTime. While TV-INR achieves this via an encoder–decoder architecture, the empirical gains over TimeFlow appear limited, especially for longer sequences (see Table 1 and Table 2). Moreover, although the authors suggest that per-sample optimization is computationally costly, Tables 21 and 22 show similar inference times for TV-INRs and TimeFlow.
3. The paper can be overly dense, and a few conceptual shortcuts are noticeable — for instance, Dupont et al. (2022) use CAVIA rather than MAML for meta-learning. There’s also a slight inconsistency between the statement “each requires per-instance optimization and may underperform in data-scarce scenarios” and the timing results shown in Tables 21–22.

**Questions:**

1. Claim #3 argues that TV-INR generalizes better when train and test settings are misaligned. This is supported by Table 12, where TimeFlow diverges under mismatched observation ratios (a known weakness of meta-learning). However, a simple meta-learning workaround is to simulate multiple missingness scenarios during training (e.g., sampling a random observation rate ∈ [0, 1] per batch). I would be very interested to see the results of Table 12 under this modified training.
2. I would also be curious to see the univariate experiments extended to more datasets, such as some of those available in the LOTSA archive.
3. Including simple baselines such as linear interpolation or a “repeat last day” heuristic could improve interpretability and help contextualize performance gains.
4. Finally, and perhaps most importantly, I strongly encourage the authors to explicitly leverage the probabilistic nature of their framework in experiments — for instance, by demonstrating quantile estimation — to reinforce the central message of the paper.

---

### Official Review · Reviewer_2bhe · 2025-10-30

**Soundness:** 2
**Presentation:** 2
**Contribution:** 2
**Rating:** 2
**Confidence:** 5

**Summary:**

This paper introduces Temporal Variational Implicit Neural Representations (TV-INRs), which is a probabilistic framework designed for individualized imputation and forecasting of multivariate time series. In the framework, the information of temporal stamps, feature vectors, and static covariates are used to learn the representation. The relevant validation experiential results are provided.

**Strengths:**

1) The idea of including the information of temporal stamps, feature vectors, and static covariates seems reasonable.
2) The proposed method is tested on different time series datasets and compared against different benchmarks.
3) Relevant ablation study results are given in the paper.

**Weaknesses:**

1)	The so-called static covariates seem to provide conditional important information for the generative model. But the essential technical details regarding the static covariates for different data types are missing in the paper.
2)	The authors proposed to use the temporal stamps information to model the sequential dependency, which is similar to Neural ODEs and diffusion models. Is this approach more suitable for modeling continuous systems than discrete systems? How is it compared to the position embedding used in Transformers?
3)	Please provide a detailed derivation of the evidence lower bound (ELBO) in Eq (3).
4)	Please explain how to compute the KL divergence between two learnable distributions in Eq (3) in detail.
5)	Since the proposed method is probabilistic, it should be evaluated using metrics such as CRPS.

**Questions:**

The proposed model is based on a VAE structure. How does the variational gap effect the model’s performance? This is not an issue for diffusion-based models.

---

### Official Review · Reviewer_iPyd · 2025-10-31

**Soundness:** 3
**Presentation:** 3
**Contribution:** 3
**Rating:** 2
**Confidence:** 4

**Summary:**

This paper presents Temporal Variational Implicit Neural Representations (TV-INRs), a probabilistic framework for modeling irregularly-sampled multivariate time series. The core idea is to integrate Implicit Neural Representations (INRs) with latent variable modeling to handle uncertainty and variability in time series data. TV-INRs learn a distribution over continuous time functions conditioned on each individual sequence’s context (including any static covariates). In contrast to many existing approaches that require heavy per-sequence optimization (fine-tuning a model to each time series or using meta-learning), TV-INRs can make accurate individualized predictions in a single forward pass. Experiments on real-world datasets  show that this method achieves competitive or superior accuracy in imputing missing values and forecasting future values, especially in low-data regimes.

**Strengths:**

- This paper is well written and easy to follow. Especially, clear notations make me easy to read this paper.
- This paper includes extensive amounts of experiments in various experimental settings and clearly states the experimental details in the appendix.
- It is interesting to see the extension of INR for time series to probabilistic modeling.

**Weaknesses:**

- **Limited novelty**:  The idea for learning continuous function is very nice for learning irregularly sampled time-series data so many researchers have been adapted this method for time-series modeling as the author mentioned in related work section. And most of techniques employed in this work is not new. There is very few technical contribution except learning "Probabilistic model" using INR. I did not find any novel engineering in loss function, neural network architecture, preprocessing and experiments, graphical model.. etc. Given that experimental results are not very strong, this mix of existing methods cannot get a good score on novelty section.

- **Omit important baselines**: Neural Differential Equation based methods(e.g. neural ODE) and Neural Processes family should be inlcuded in the baselines as the authors introduce probabilistic continuous time model.

- **Unfair comparison**: Comparing this with gradient-based meta-learning approaches which requires per-sample optimization during inference and claiming for "efficiency" seems unfair. I think the proper baselines should be other probabilistic model for time-series modeling because meta-learning in inference time is specific for some INR models like Le Naour et al., 2024.

- **Performance on large scale experiments**: This method shows less competitive performance on high-data regime compared to low-data regime. This will limit the adoption of this method for practitioners.

**Questions:**

- Why do you not include uncertainty related (e.g., calibration, visualization for ood samples) results? It is natural for probabilistic model for INR to stress how good at measuring the functional uncertainty. Otherwise, why should we use this model?

---

### Official Review · Reviewer_iqXV · 2025-11-06

**Soundness:** 2
**Presentation:** 3
**Contribution:** 3
**Rating:** 4
**Confidence:** 4

**Summary:**

This paper presents a new framework for time series imputation and forecasting. The proposed model combines implicit neural representations with variational autoencoders to model time series in a continuous manner. The approach combines a Transformer encoder, variational autoencoder framework, and
hypernetwork-generated implicit neural representations to model  time-continuous generator functions conditioned on partial observations and static covariates. The approach differs from previous work through the use of amortized inference to learn distributions over INR parameters (using hypernetworks)  to
eliminate per-sample gradient-based optimization during inference. Experiments are conducted on classical imputation and forecasting problems to demonstrate the framework's effectiveness compared to state-of-the-art algorithms, along with ablation studies.

**Strengths:**

* Well-founded probabilistic framework that rigorously combines VAE principles, amortized inference, and INRs
* The model demonstrates  superior results on  sparse observations
* The VAE implementation with conditional priors is technically sound, and the integration of Transformer encoders with hypernetwork-generated INRs is properly implemented
* The covariates variant  demonstrates that incorporating static covariates can improve performance at extreme sparsity thought  gains are inconsistent across datasets.

**Weaknesses:**

* The presented work is architecturally very similar to HyperTime (Fons et al., 2022), which combines set encoders, hypernetworks, and INRs for time series. I think the
main differences in TV-INRs are: (1) using a Transformer encoder instead of DeepSets, and (2) adopting an explicit VAE framework with prior/posterior distributions instead of deterministic latent representations.
There is very little discussion of how this work positions itself relative to HyperTime, which is detrimental to understanding the actual contribution. Furthermore, the experiments do not include any comparison with HyperTime, with the authors justifying this omission by the lack of publicly available code. However, given the conceptual proximity of the two approaches, a re-implementation was essential to validate the contributions.

* One of the paper's main arguments is computational efficiency and scalability.  However, the inference time results in Tables 21-22 (Appendix A.11) contradict these claims.The measurements show that TV-INRs has comparable inference time to TimeFlow for short sequences, but becomes significantly slower for long sequences. This degradation is explained by the complexity of the Transformer encoder, whose cost exceeds the savings from avoiding TimeFlow's gradient-based optimization. It is problematic that these results, which significantly undermine the efficiency claims, are relegated to the appendix without discussion in the
main paper.

* While the paper includes experimental evaluation on 6 datasets (4 univariate, 2 multivariate), the coverage for comparisons with TimeFlow and other classical SOTA papers is limited. The paper tests only 3 usual univariate datasets(Electricity, Traffic, Solar) without considering  ETTh1/h2, ETTm1/m2, Weather, Exchange-Rate, and ILI. With only 3 overlapping datasets, it is difficult to determine whether the observed patterns (TV-INRs better in low-data, TimeFlow better in high-data) generalize broadly. Evaluation on the complete set of standard benchmarks would have provided a more robust and comprehensive view of the proposed approach's capabilities and limitations.

**Questions:**

* One of TimeFlow’s merits is its ability to perform imputation and forecasting simultaneously. Can TV-INRs handle joint imputation + forecasting?

* Why select only 3 univariate datasets  among the ~10 usual benchmarks ?

---

### Note · Authors · 2025-11-14

I have read and agree with the venue's withdrawal policy on behalf of myself and my co-authors.